# Mapping aquifer salinity gradients and effects of oil field produced water disposal using geophysical logs: Elk Hills, Buena Vista and Coles Levee Oil Fields, San Joaquin Valley, California

**Janice M. Gillespie**[1]*, **Michael J. Stephens**[2], **Will Chang**[3], **John G. Warden**[1]

**1** U.S. Geological Survey, San Diego, California, United States of America, **2** U.S. Geological Survey, California Water Science Center, Sacramento, California, United States of America, **3** Hypergradient LLC, Berkeley, California, United States of America

* jmgillespie@usgs.gov

**Data Availability Statement:** All data files are available from Stephens, M.J., Gillespie, J.M., Warden, J.G., and Chang, W., 2021, Geophysical,

## Abstract

The effects of oil and gas production on adjacent groundwater quality are becoming a concern in many areas of the United States. As a result, it has become increasingly important to identify which aquifers require monitoring and protection. In this study, we map the extent of groundwater with less than 10,000 mg/L TDS both laterally and vertically near the Elk Hills, Buena Vista and Coles Levee Oil Fields in the San Joaquin Valley, California and note evidence of effects of produced water disposal on salinity within the Tulare aquifer. Subsurface maps showing the depth at which groundwater salinity is less than 10,000 mg/L (or Base 10K) in the Tulare aquifer are generated using geophysical logs and verified by comparison to water sample analyses. The depth to Base 10K ranges from 240 m (800 ft) in Elk Hills to 800 m (2650 ft) in the adjacent Buena Vista syncline and is 670 m (2,200 ft) deep in the Coles Levee area to the east. Log-calculated salinities show a relatively smooth increase with depth prior to disposal activities whereas salinities calculated from logs collected near and after disposal activities show a more variable salinity profile with depth. The effect of produced water injection is represented by log resistivity profiles that change from low resistivity at the base of the sand to higher resistivity near the top due to density differences between the saline produced water and the brackish groundwater within each sand. Continued post-disposal logging in new wells in the 18G disposal area on the south flank of Elk Hills shows that injected water has migrated approximately 1,200 m (4,000 ft) downdip (south) over a period of 20 years since the inception of disposal activity.

## Introduction

As oil and gas production has increased in many areas of the United States (US) due to the introduction of new technologies such as directional drilling and hydraulic fracturing, concerns

geological, hydrological, and geochemical data for aquifer salinity mapping in the Elk Hills Oil Field area, Kern County, California: U.S. Geological Survey data release, https://doi.org/10.5066/P9KWNEFW database. This is part of the USGS ScienceBase Catalog.

**Funding:** This work was primarily funded by the California State Water Resources Control Board Oil and Gas Regional Monitoring Program and supplemental US Geological Survey Cooperative Matching Funds. The funders had no role in study design, data collection and analysis, decision to publish, or preparation of the manuscript.

**Competing interests:** The authors have declared that no competing interests exist.

have arisen over the potential effects of these activities on groundwater quality. California Senate Bill 4 (SB 4 statutes of 2013) authorized the California State Water Resources Control Board (Water Board) to implement a program to monitor water quality in areas of well stimulation beginning in 2015. California has a long-term interest in better understanding potential interactions between oil field operations and adjacent groundwater quality and, after passage of SB4, the Water Board set up the Regional Monitoring Program (RMP) and invited the U.S. Geological Survey (USGS) to serve as the technical lead. The RMP includes assessing potential impacts to groundwater resources associated with well stimulation (hydraulic fracturing), enhanced recovery (water and steam flooding), and disposal of produced water (water brought to the surface along with oil and gas production) by underground injection or surface sumps [1].

Brackish groundwater resources with total dissolved solids (TDS) of 1,000 to 10,000 milligrams per liter (mg/L) are increasingly considered for domestic and industrial use because these resources can be treated for a lower cost than desalination of seawater [2–5]. In general, aquifers containing water with less than 10,000 mg/L TDS are classified as Underground Sources of Drinking Water (USDW) (https://www.epa.gov/uic/epa-oversight-californias-underground-injection-control-uic-program, accessed 11/2/2021). Specific aquifer zones may be exempt from this level of protection when they do not currently, or are not expected to, serve as a source of drinking water (https://www.epa.gov/uic/epa-oversight-californias-underground-injection-control-uic-program; accessed 11/2/2021). Oil field operations, including water and steam flooding and injection of waste, can be permitted in these exempt areas even if salinities are below 10,000 mg/L TDS. California has been reviewing a number of specific aquifer exemption applications, as well as updating the regulatory program governing underground injection in oil fields (https://www.conservation.ca.gov/calgem/Documents/UIC%20regulations/UIC%20Final%20Statement%20of%20Reasons.pdf, accessed 5/22/2021).

Larger scale regional studies have estimated the lateral and vertical extent of groundwater with less than 2,000 mg/L TDS in the Central Valley of California [6], and recent work in the San Joaquin Valley (SJV) has determined the lateral and vertical extent of groundwater with less than 10,000 mg/L TDS regionally [7] and more specifically near the Lost Hills and Belridge Oil Fields [8]. However, the extent of groundwater resources with less than 10,000 mg/L TDS has not been sufficiently mapped in other parts of the SJV including within and in proximity to the Elk Hills, Buena Vista, and Coles Levee Oil Fields. To address this knowledge gap, this study makes extensive use of borehole geophysical log analysis to determine the depth to the base of groundwater containing less than 10,000 mg/L TDS, thus delineating aquifers with than 10,000 mg/L TDS within and in proximity to the Elk Hills, Buena Vista, and Coles Levee Oil Fields. To assess the effects of produced water disposal, this study uses open-hole geophysical logs collected from adjacent wells drilled at different times to map changes in groundwater resistivity through time, which may be related to saline produced water disposal in injection wells or surface ponds within and overlying the primary aquifer in the area—the Tulare Formation. This approach can be used to map the salinity distribution of groundwater within the aquifer as well as the presence of relatively higher salinity oil field produced waters in groundwater in proximity to oil fields and to determine movement of disposal water over time. The use of geophysical logs to map groundwater salinity is well established and has been used in areas of Texas [9], Utah [10] and, on a more regional scale, in the San Joaquin basin [6, 7, 11].

The Elk Hills-Buena Vista-Coles Levee study area lies in the southern San Joaquin Valley (SJV) in Kern County, California. The area is located approximately seven miles southwest of Bakersfield, California and contains four large oil fields: Elk Hills, Buena Vista, North Coles Levee and South Coles Levee (Fig 1). The Elk Hills Oil Field (formerly Naval Petroleum Reserve No.1) has produced over one billion barrels (160,000,000 m$^3$) of oil since its discovery in 1911 and is one of the 10 largest oil fields in California, ranking 7$^{th}$ in oil production and

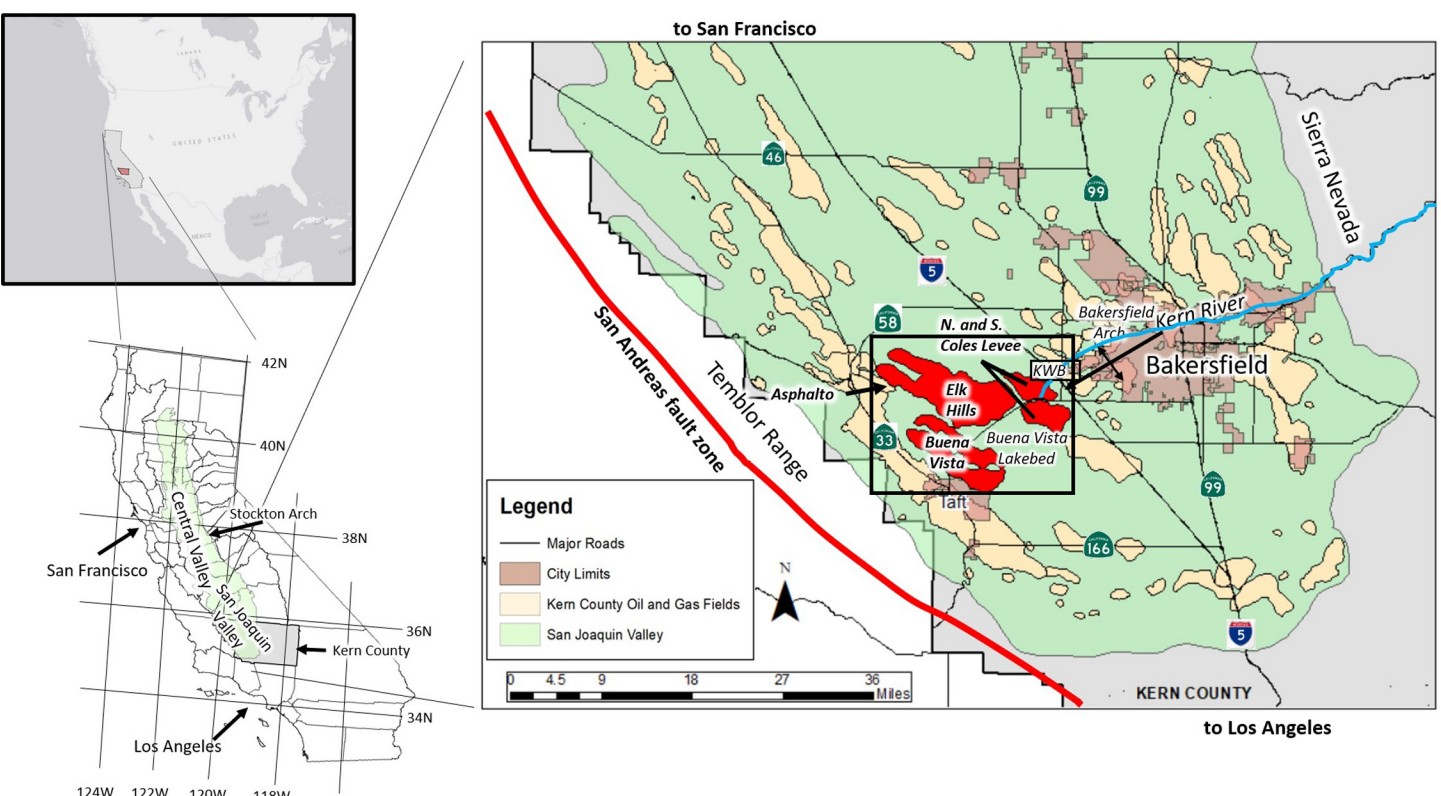

**Fig 1. Location of the study area.** The study area is outlined in black box showing the producing areas of oil fields in the study area highlighted in red.

first in associated gas production in California in 2018 [12]. The Buena Vista Oil Field (former Naval Petroleum Reserve No. 2) ranked a distant second in associated gas production in the state [12]. The Coles Levee Oil Fields are adjacent to the Kern Water Bank (KWB), an area in which surface water is stored underground in wet years for later use during drought years.

The oil in the study area ranges from 18 to 57° API (American Petroleum Institute) (except for a small amount of 10° API gravity crude in the Tulare Formation in western Elk Hills) [13]. It is produced from a variety of shallow and deep marine Miocene and Pliocene sandstone and shale reservoirs—primarily the Stevens sands of the Monterey Formation and the San Joaquin and Etchegoin Formations (Fig 2)-at depths ranging from 610 to 3,040 m (2,000 to 10,000 ft). Dry gas is produced from Pliocene sands in the San Joaquin Formation (Dry Gas Zone) at depths ranging from 305 to 1825 m (1,000 to 6,000 ft) [13]. Water quality in the eastern part of the study area, near the Coles Levee Oil Fields, contains fresh groundwater heavily used as a source of irrigation and intermittently used for domestic supply.

## Geologic setting

The SJV forms the southern half of the Central Valley of California (Fig 1). Intrusive igneous and metamorphic rocks of the Sierra Nevada lie to the east and south and, to the west and south, the Temblor Range within the Coastal Ranges contains igneous, metamorphic and sedimentary rocks. The broad, low amplitude uplift known as the Stockton Arch lies east of San Francisco and marks the northern geologic boundary of the SJV [15].

The Central Valley developed as a forearc basin adjacent to an east-dipping subduction zone [15]. During the early Miocene, the East Pacific Rise encountered the trench offshore

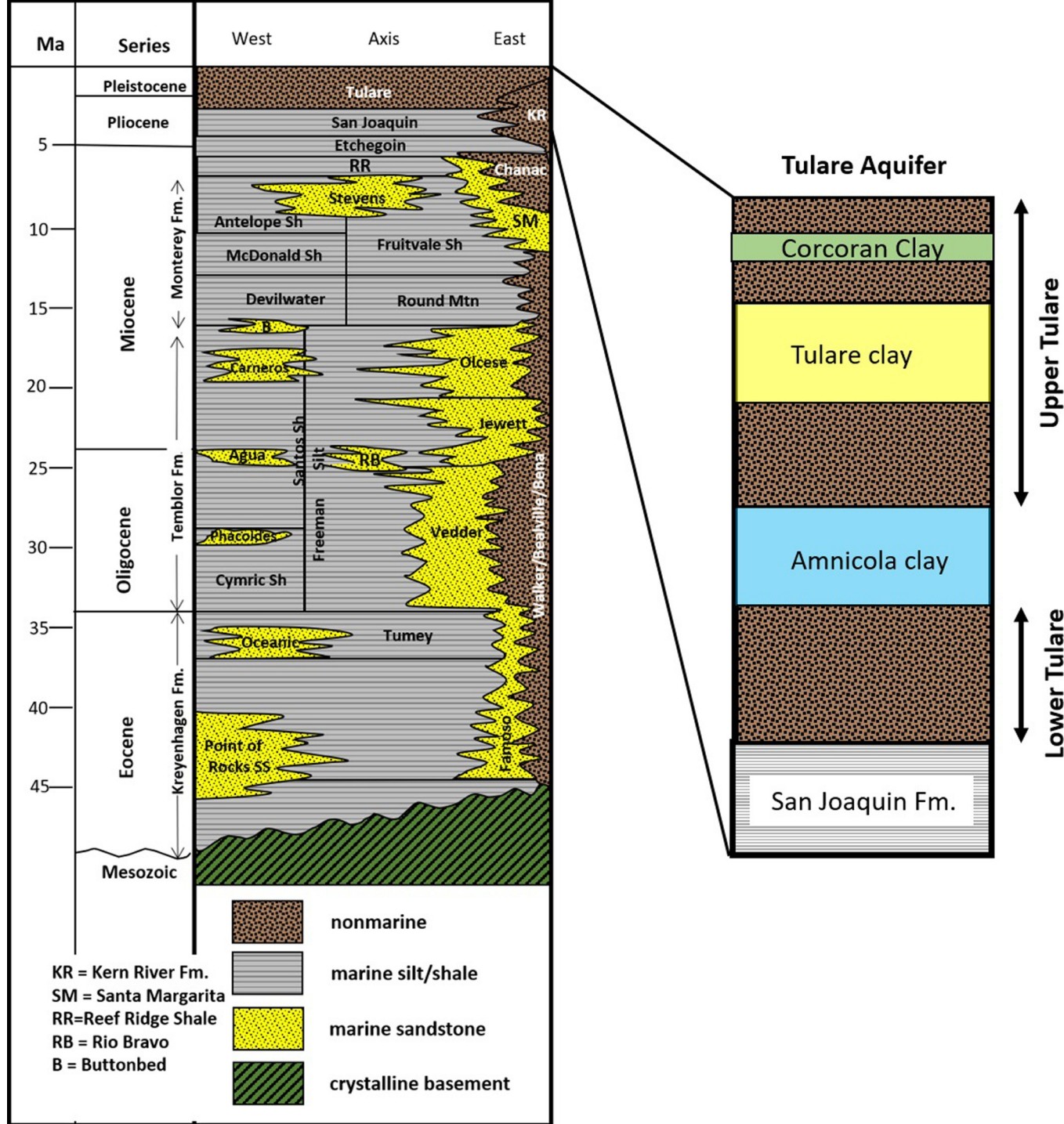

**Fig 2. Stratigraphy in the study area.** The study area lies near the basin axis. The stratigraphy of the Tulare aquifer is shown in greater detail. Modified from Scheirer and Magoon (2007) [14].

present-day southern California and a transform basin margin was established, creating today's San Andreas Fault [16], which lies near the western margin of the SJV (Fig 1). Movement along the San Andreas Fault during the Miocene often cut off the southernmost San

Joaquin Valley from open ocean circulation [15, 17], forming a bottom layer of anoxic sea water in which organic-rich shales of the Monterey Formation (Fig 2) formed in a deep ocean setting [17]. These shales ultimately became the source of much of the oil in the associated turbidite sands and overlying formations [17, 18].

Over time, the southern San Joaquin basin experienced increased rates of sediment input and shallow marine sediments of the Etchegoin and San Joaquin Formations were deposited above the deep water shales and turbidites of the Monterey Formation and Reef Ridge Shale [19]. As the basin continued to fill with sediments, nonmarine conditions prevailed, and nonmarine deposits of the Pleistocene age Tulare Formation replaced the shallow marine deposition present during Pliocene time [15] (Fig 2).

The southern part of the SJV is tectonically active. Neogene transpressional forces generated along the transform margin formed an east-northeast vergent fold and thrust belt east of the San Andreas Fault along the western boundary of the southern SJV [20]. The resulting folds and faults created numerous traps for large pools of oil. The Elk Hills, Buena Vista and Coles Levee anticlines form the eastern part of this fold-thrust belt and lie just west of the present-day topographic basin axis. These anticlines separate the southern SJV into two sub-basins: the Buttonwillow sub-basin to the north and the Maricopa sub-basin to the south [20] (Fig 3).

The structure map in Fig 3 (see Methods section for details regarding map construction and Stephens et al. [23] for information on formation picks) shows the elevation of the base of the Tulare Formation with respect to sea level and illustrates the folding of the strata in the study area. The Elk Hills structure reaches a topographic elevation of 374 m (1,227 ft) and lies approximately 305 m (1,000 ft) above the valley floor. In the subsurface at the depth of the Monterey Formation, the Elk Hills Oil Field consists of three anticlines underlain by thrust faults: the 31S, 29R and Northwest Stevens anticlines [15, 24]. However, only a single doubly plunging anticline is visible above the Monterey Formation in the Pleistocene Tulare Formation [25] (Fig 3). In contrast to the thrust faulting in the deeper zones, the shallow formations are affected by a series of northeast-trending listric normal faults. The dip of these faults decreases with depth and they sole out (become horizontal) above the Monterey Formation [26, 27]. Four of these faults were initially mapped at the surface [22] and are shown in Fig 3. However, many similar faults are mappable in the subsurface in both eastern Elk Hills and the Coles Levee Oil Fields [27–29].

The east-plunging nose of the Elk Hills anticline bifurcates into the southeast-plunging North and South Coles Levee anticlines in the eastern part of the study area. These anticlines are not visible as surface features and were discovered during the advent of seismic prospecting in the area in the 1930's. The topographic basin axis lies within the eastern part of the Coles Levee Oil Fields (Fig 3) and coincides with U.S. Interstate 5 (Fig 1). East of the Coles Levee Oil Fields, the land surface gently rises into the foothills of the Sierra Nevada along a broad, southwest-plunging structural high known as the Bakersfield Arch (Figs 1 and 3). The Bakersfield Arch hosts several small oil fields along its crest and upon its flanks east of the study area.

Topographically, the Buena Vista Oil Field is located in a series of rolling hills (the Buena Vista Hills) up to 390 m (1,280 ft) high (about 260 m (850 ft) above the valley floor) formed by an asymmetrical anticline that trends subparallel to the Elk Hills structure. It is separated from Elk Hills by the Buena Vista Valley—the surface expression of a southeast plunging synclinal feature (Fig 3) which contains the Buena Vista lakebed (Fig 1). Along the crest of the structure are two high areas: the East and West domes [30]. The plunge of the Buena Vista anticline southeast of the East Dome continues outside the oil field administrative boundary forming the Buena Vista Nose area (BV Nose, Fig 3) which contains several wells producing from sandstones of the Monterey Formation [31].

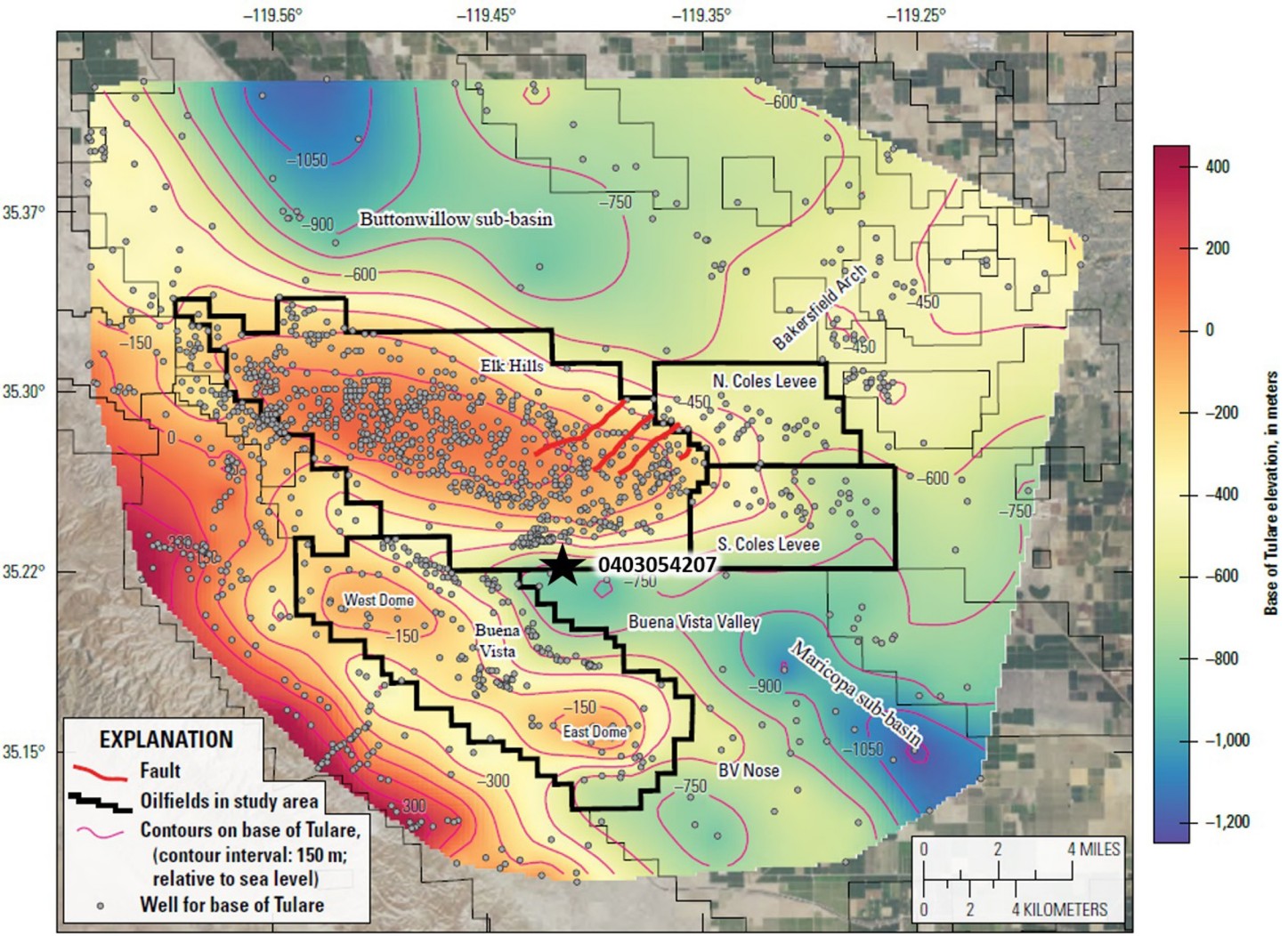

**Fig 3. Structural contour map showing the elevation of the base of the Tulare Formation with respect to sea level.** Pertinent structural and geographic features are labeled. Wells shown on the map are those used to generate the contours. Star shows well 0402938955 in which the contact between the Tulare Formation and overlying alluvium was defined by Milliken (1992) [21]. Contour interval = 150 m. Faults mapped at the surface in eastern Elk Hills [22] are shown in red. Heavy black outlines show oilfield administrative boundaries in the study area. Air photo base from https://basemap.nationalmap.gov/arcgis/rest/services/USGSImageryOnly/MapServer/.

## Hydrogeology

The west side of the SJV lies in the rain shadow of the Coast Ranges which separate the SJV from the Pacific Ocean. Consequently, annual rainfall in this area averages only about 13 to 23 centimeters (cm) (5 to 9 inches (in)) [32]. East of the study area, storm systems must rise to cross the larger barrier of the Sierra Nevada and rainfall on the western flanks of the Sierra Nevada is much higher, ranging from 15 centimeters per year (cm/yr) (6 inches per year (in/yr)) at the foot of the Sierra Nevada in Bakersfield [33] to over 127 cm/yr (50 in/yr) in the mountainous areas of the southern Sierra Nevada [34]. In addition, water derived from snow-melt in the high Sierra Nevada is carried to the eastern part of the SJV by a series of west-flowing streams which, upon entering the basin, change from gaining to losing reaches and recharge the underlying aquifer. Water from northern California is also imported into the area

via the California Aqueduct (State Water Project) and Friant-Kern Canal (Central Valley Project) to augment natural supplies.

The main west-flowing stream in the study area is the Kern River (Fig 1) which flows from the southern Sierra Nevada through the city of Bakersfield and, when winter snows in the Sierra Nevada are heavy, may travel all the way to the basin axis in the eastern part of the Coles Levee Oil Fields. The area is a closed hydrologic system and the river's flows do not normally reach the Pacific Ocean. Prior to construction of the Isabella Dam upstream in the Sierra Nevada, the Kern River's natural flow was southwestward into the Buena Vista lakebed (Fig 1) which lies in the southeast part of the study area [35]. Today, excess flows are diverted into the California Aqueduct, which runs near the eastern boundary of the study area, or (since 1995) into constructed ponds within the Kern Water Bank (KWB) [36]. The KWB straddles the eastern boundary of the study area (Fig 1). Storage of the excess Kern River flows in the KWB ponds allows the water to seep into the underlying aquifer from which it can be withdrawn during drought years.

The major aquifer in the area is in the Pleistocene Tulare Formation (Fig 2) and overlying alluvium which are composed of fluvial sands and gravels and lacustrine clays [37]. The Tulare aquifer forms the southern portion of the Central Valley regional aquifer system. This regional aquifer system is well-documented and has been the subject of numerous hydrogeologic and groundwater flow modeling studies [38, 39]. In the study area, the thickness of the Tulare Formation and overlying alluvium ranges from about 1,200 m (4,200 ft) in the Buttonwillow and Maricopa sub-basins in the northwestern and southeastern parts of the study area to 305 m (1,000 ft) over the crest of the Elk Hills anticline where it is largely unsaturated. The Tulare Formation crops out in the Elk Hills and Buena Vista Oil Fields. It was extensively mapped at the surface of Elk Hills [22]. Milliken [21] mapped The Tulare Formation and overlying alluvium at the surface at the Elk Hills Oil Field and noted that the alluvium is composed of reworked Tulare Formation strata on the south flank of the Elk Hills Oil Field and that, while the Tulare Formation dips 9–20 degrees southeastward on the south flank of the Elk Hills Oil Field, the alluvium is not deformed and unconformably overlies the outcrops of the Tulare Formation. The contact between the Tulare Formation and the overlying alluvium was placed at a depth of 144 m (473 ft) in well 0402938955 in the Buena Vista Valley (Fig 3) but this contact is highly subjective in the subsurface [21]—for this reason, the Tulare Formation/alluvium contact was not mapped in this study.

The Tulare Formation contains three major lacustrine clay layers in the study area (Fig 2). However, these clays are only well developed in the western part of the study area. ICF [36] in their environmental impact report on the Kern Water Bank (which covers much of the North Coles Levee Oil Field), noted that there are no laterally extensive clay deposits (eg. the Corcoran Clay) in the eastern part of the study area in the Coles Levee Oil Fields and that it is very difficult to find any single deposit that can be correlated with confidence across the Kern River alluvial fan. They note that this has resulted in a leaky aquifer as evidenced by hydraulic head data from monitoring wells at the Kern Water Bank.

The oldest clay is known as the Amnicola clay (named for freshwater gastropods of *Amnicola sp.* that are common within the clay [26]), and it occurs throughout the study area with the exception of the eastern Coles Levee area where a facies change causes the clay to be replaced by sand layers deposited by the Kern River. Miller [40] conducted a comprehensive study of the Tulare Formation in the southern SJV based on outcrops, seismic data and well logs. The Amnicola clay is thought to correlate to Miller's [40] orange seismic horizon with an age of 2.2 Ma [41]. For the purpose of this study, upper and lower intervals of the Tulare Formation are defined relative to the Amnicola clay.

The middle clay is known as the Tulare clay [21, 42] and is also called the Basal Alluvial clay [43]. It occurs only around the margins of the Elk Hills and Buena Vista anticlines and is exceptionally thick (over 305 m (1,000 ft)) below the Buena Vista lakebed in the southeast. A facies change occurs in the Coles Levee oil fields where the Tulare clay is replaced by sand layers of the Kern River alluvial fan. A limestone layer has been mapped within the upper part of the Tulare Formation at Elk Hills [22], and the Tulare clay interval crops out on the south flank of the Elk Hills anticline in the vicinity of the mapped limestone [21]. This limestone and clay interval is thought to continue into the subsurface as the Tulare clay [41]. Early maps of the upper Tulare Formation lacustrine limestone layers that appear to correlate to the Tulare clay show them at an elevation of at least 275 m (900 ft)—over 150 m (500 ft) above the valley floor [22], and seismic correlations suggest that the Tulare clay correlates to Miller's [40] neon (1.4 Ma) or yellow (1.0 Ma) seismic horizons [41]. The occurrence of the lacustrine clay layer and associated limestone high above the valley floor on the Elk Hills anticline, suggests that the Elk Hills were probably not as topographically high during deposition of the Tulare clay as they are today, and that the topographic expression of the structure may be as recent as 1.0 Ma [41]. This uplift may have caused the groundwater within the Tulare Formation to drain into the lower topographic regions in the Buttonwillow and Maricopa sub-basins creating the thick (about 300 m (1,000 ft)) vadose zone observed within the aquifer over the crest of the Elk Hills anticline. This thick vadose zone is currently being targeted for produced water disposal at Elk Hills.

The Corcoran Clay Member of the Tulare Formation (Corcoran Clay) [44] or E-clay of Croft [45] is a lacustrine clay which, in some studies, is interpreted to lie above the Tulare clay [21, 43] and is present in some oil and water well logs in the study area [39, 45]. While early studies assumed that the Tulare aquifer system was comprised of an upper unconfined and lower confined aquifer separated by the Corcoran Clay, more recent modeling studies treat the aquifer system as a single heterogeneous aquifer [38]. The Corcoran Clay is not known to crop out on the flanks of the Elk Hills structure and is not apparent in well logs at Elk Hills [21]. A recent study suggests that the Corcoran Clay likely onlapped the emerging topography of the Elk Hills structure and did not cover the structure [41].

Natural groundwater in the aquifer contains TDS of 4100 to 12,600 mg/L and is too poor quality to be used for domestic or agricultural purposes without extensive treatment except for the eastern part of the area over the Coles Levee Oil Fields where recharge from the Kern River is more frequent and abundant. Water in the Kern River is a calcium-sodium-bicarbonate (Ca-Na-HCO$_3$) type, and the TDS measured in a river water sample collected near Bakersfield in 1952–55 (when there was little regulation of flow by the Isabella Dam) averaged 69 mg/L [46]. This area contains many wells used for domestic and agricultural purposes and hosts the western portion of the Kern Water Bank.

In contrast, little recharge occurs in the western part of the study area which lies within the rain shadow of the Temblor Range. Streams draining the east side of the Coast Ranges are generally ephemeral and flow mainly in winter and spring. A sample collected from an ephemeral stream near the eastern end of the boundary between Elk Hills and Buena Vista Oil Fields (T31S, R24E, sec. 19) was of Na-SO$_4$ type with TDS of 1,030 mg/L [46].

Water types for groundwater samples from Tulare aquifer water wells, typically collected from depths less than 305 m (1,000 ft), can generally be divided into three groups in the study area: (1) "East-side" Ca-HCO$_3$ or Ca-Na-HCO$_3$ groundwater of low TDS (less than 300 mg/L) originating from recharge of the Kern River and common in deposits of the Kern River alluvial fan, (2) "West-side" Na-SO$_4$ or Na-Cl-SO$_4$ groundwater of moderate TDS (up to about 5,000 mg/L), and (3) "axial trough" groundwater of variable composition common south and southwest of the Kern River alluvial fan and in the Buena Vista Lake bed [32, 46, 47]. These three

typical compositions in relatively shallow groundwater may be influenced by rock/water interactions, managed aquifer recharge, agricultural activities, and historical oil field water disposal activities in the last century in some areas [32, 46–48].

## Oil field operations and water disposal

It is common for water oil ratios (WOR—barrels of water produced per barrel of oil produced) to increase over the life of an oil field. As oil is removed, groundwater often comes in to take its place. In 2018, the WOR for Elk Hills Oil Field was 15, Buena Vista Oil Field was 38, and North and South Coles Levee Oil Fields each had a WOR of three (Table 1). Some produced water is used for enhanced oil recovery (EOR) practices such as water flooding. The remaining water is disposed of in several ways. Historically, water disposal in the SJV was often accomplished by spreading the produced water on the ground, usually in surface ponds or dry stream beds, and the water was allowed to evaporate or percolate into the underlying alluvium [49].

A group of both state and federal legislative bills, including the 1969 Porter Cologne Water Quality Act, the 1972 Clean Water Act, and the 1974 Safe Drinking Water Act, provided government agencies authority to regulate surface water and groundwater degradation [49]. As a result, many surface disposal ponds were abandoned and produced water disposal began to move to injection via water disposal wells into non-oil producing zones. The federal Underground Injection Control (UIC) program established nationwide requirements for the protection of aquifers containing water with less than 10,000 mg/L TDS and meeting other use criteria [50].

Currently, most of the produced water in the Elk Hills and Buena Vista Oil Fields is injected into oil reservoirs for EOR purposes. The rest of the produced water is injected into water disposal wells, primarily into the Tulare Formation, by permit in state and federal approved exempt aquifer zones. Produced water disposal (WD) by injection into the Tulare Formation (and, in the case of the Coles Levee Oil Fields, into the lower Tulare and San Joaquin Formations) began in 1980 in the Elk Hills Oil Field, 1972 in the Buena Vista Oil Field and 1963 in the Coles Levee Oil Fields. Since 1977, 62.5 million $m^3$ (393 million barrels) of produced water has been injected into the Tulare Formation at the Buena Vista Oil Field for disposal (Fig 4A) [51]. The total injection for disposal at Elk Hills Oil Field is much higher—approximately 222.5 million $m^3$ (1.4 billion barrels) (total for the 18G, 27R and 23/25Z areas—this amount includes 23 wells on the northeastern flank of the Asphalto Oil Field (Fig 1)—the 23/25Z WD area in western Elk Hills Oil Field straddles the boundary between the two oil fields). At the Buena Vista and Elk Hills Oil Fields, analysis of well histories on the California Geologic Energy Management Division (CalGEM) online database [51] indicate that injection has resulted in several surface expressions (Fig 4B and S1 Table)—defined in the UIC draft regulations as "a flow of fluid or material to the surface that is not through a well and is caused by injection operations".

(https://www.conservation.ca.gov/calgem/general_information/Documents/UIC_regs_workshop/Final%20Text%20of%20the%20UIC%20Regulations%20(Clean).pdf; accessed 4/14/2021). Not all well records were examined during this study to determine whether surface

**Table 1. Fluid production in cubic meters ($m^3$) and United States barrels (bbls) and gas production in cubic meters ($m^3$) and thousand cubic feet (MCF) for the oil fields within the study area in 2018.** The ratio of produced water to oil is shown in the last column. Values from California Geologic Energy Management Division (CalGEM) [12].

| oil field | oil (bbls) | oil (m3) | water (bbls) | water (m3) | gas (MCF) | gas (m3) | water-oil ratio |
|---|---|---|---|---|---|---|---|
| Buena Vista | 1,297,937 | 206,356 | 49,075,786 | 7,802,426 | 14,408,096 | 407,749,117 | 38 |
| Coles Levee, North | 82,321 | 13,088 | 228,210 | 36,283 | 102,494 | 2,900,580 | 3 |
| Coles Levee, South | 51,479 | 8,185 | 144,030 | 22,899 | 466,335 | 13,197,281 | 3 |
| Elk Hills | 8,565,920 | 1,361,872 | 129,913,797 | 20,654,640 | 89,200,374 | 2,524,370,584 | 15 |

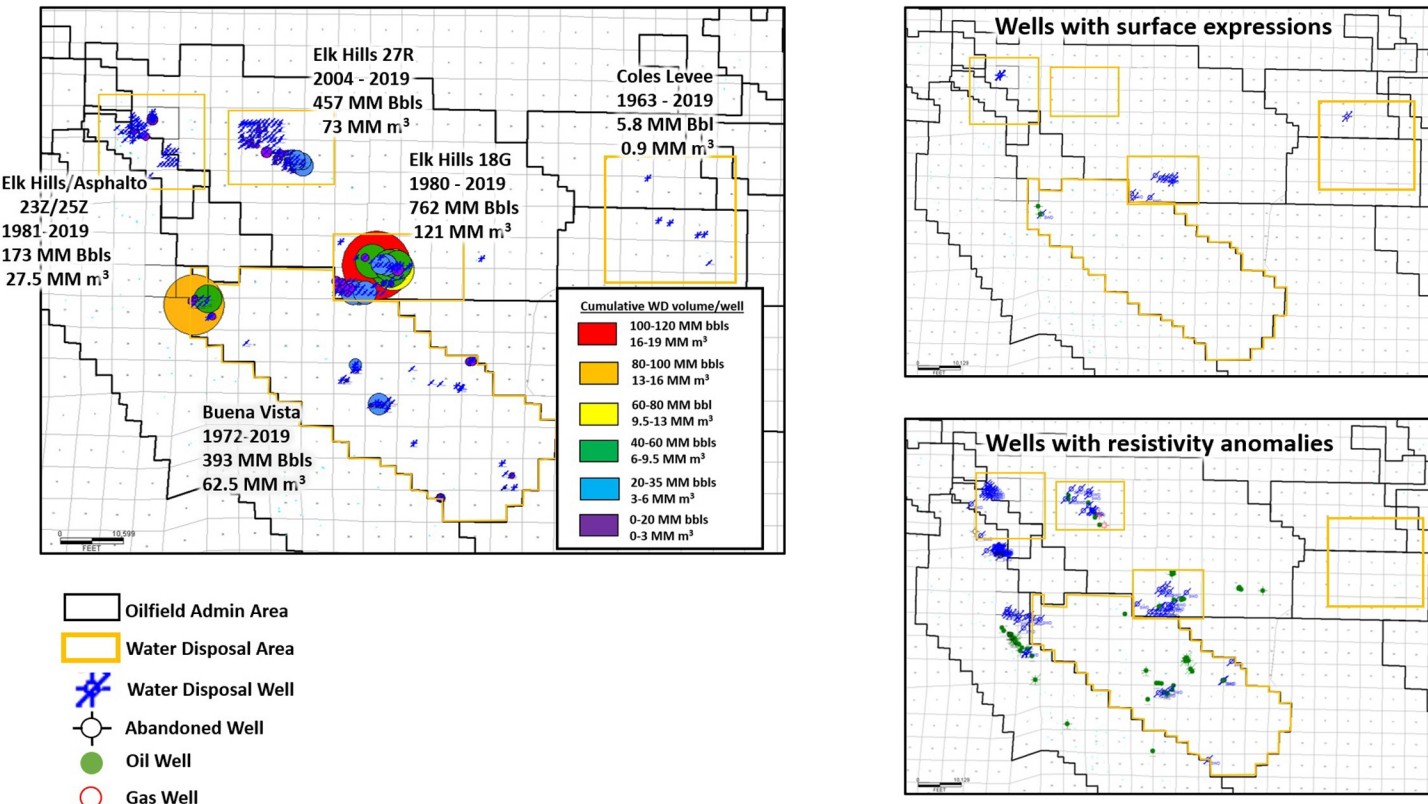

**Fig 4. Oil field produced water disposal by injection in the study area.** a) Bubble map showing volumes of produced water disposal in each water disposal well. The size and color of the circle represents the volume of injected water post-1977 (MM bbls = million barrels) [51]. The dates shown on the disposal area labels represent the dates during which disposal by injection into the Tulare Formation occurred within the oil fields. Totals for Buena Vista Oil Field and the North and South Coles Levee Oil Fields combined are shown within the area over which the volumes were totaled (outlined in orange). At the Elk Hills Oil Field, the three main disposal areas (18G, 23/25Z and 27R) are summed separately. b) Wells with surface expressions (fluids breaking through to the surface outside the casing) and c) wells with resistivity anomalies possibly related to the injection of produced water.

expressions have occurred, therefore this study represents a minimum estimate for the study area [23]. At Coles Levee, less than 954,000 m$^3$ (six million barrels) have been injected for disposal into wells completed in the lower Tulare and San Joaquin Formations since 1977. Much of the produced water in these oil fields is injected into the deeper Etchegoin Formation.

## Methods

All data used in the analysis are available from Stephens et al. [23]. The data includes information on the formation picks (base Tulare, top and bottom of the Amnicola and Tulare clays) as well as hydrologic and water quality information such as the depth to groundwater with TDS greater than 10,000 mg/L (base 10K) and the top of the water table as determined from water level measurements and density-neutron log cross over. A table containing log calculated TDS in wells used in TDS versus depth plots is also included.

## Mapping

Geophysical well log correlations were used to map the elevation of the base of the Tulare Formation (Fig 3) and its thickness (Fig 5), the net clay thickness and extent of the Amnicola clay (Fig 6A) and the gross interval thickness and extent of the Tulare clay (Fig 6B). Electric logs (resistivity and spontaneous potential (SP)) and/or gamma ray logs from 1624 wells were used

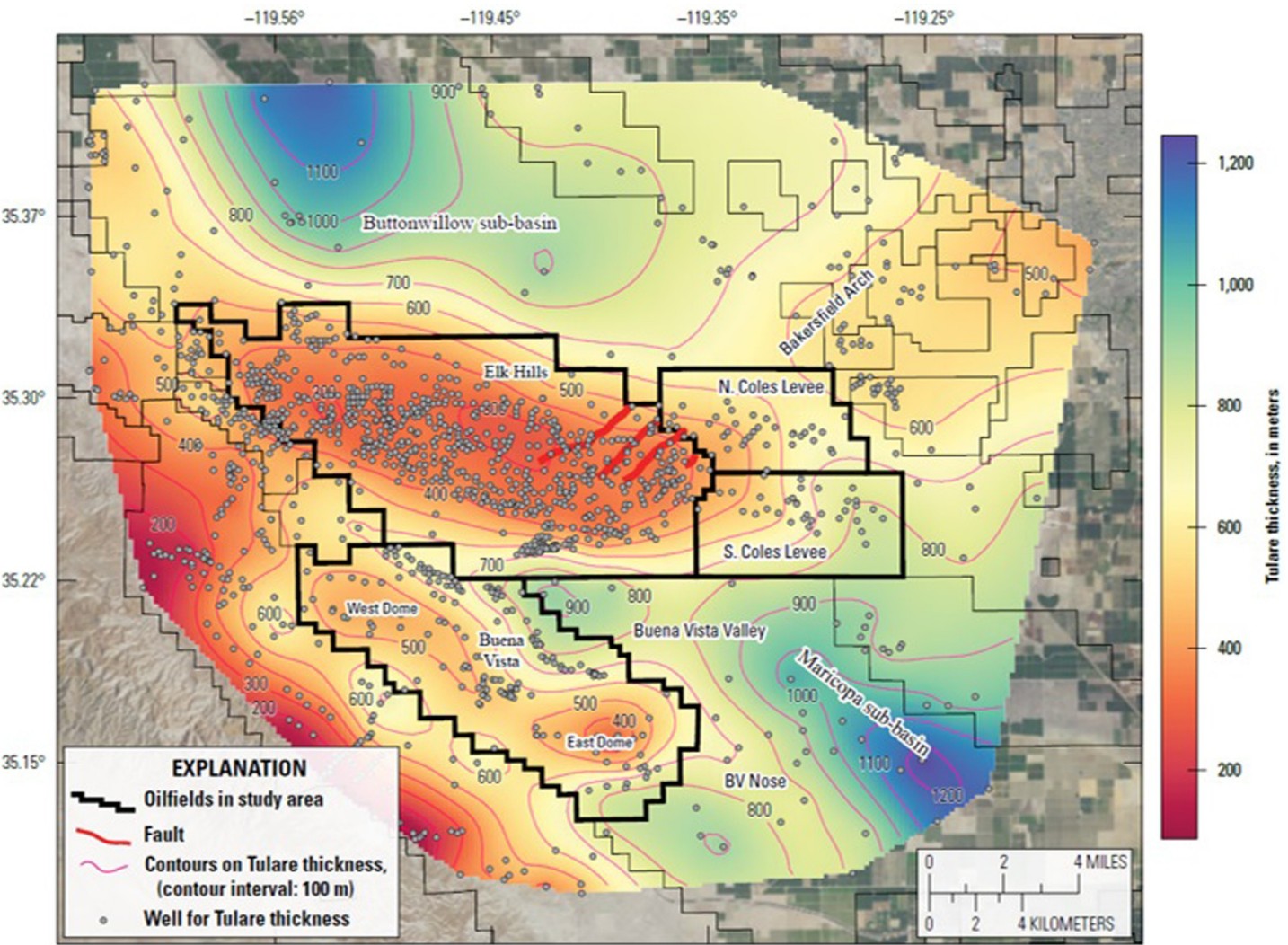

**Fig 5. Map showing thickness of the Tulare Formation in the study area.** The formation is thinner over the crests of the anticlines and thicker in the adjacent synclines. Heavy black outlines show oil field administrative boundaries. Wells shown are those used to generate the contours. Contour interval = 100 m. Air photo base from https://basemap.nationalmap.gov/arcgis/rest/services/USGSImageryOnly/MapServer/.

to identify correlatable layers. Correlation picks for each well may be found in Stephens et al. [23]. Data regarding the formation picks come from two main sources: 1) oil well histories available from CalGEM [51] and 2) aquifer exemption applications from California Resources Corporation [42, 52]. For wells where this information was not available, the base of the Tulare Formation throughout most of the study area is defined by a change from high sand content and poorly correlatable fluvial sands in the Tulare Formation to markedly lower sand content with more easily correlatable sand bodies in the underlying San Joaquin Formation. There is also a shift in the gamma ray log from higher overall gamma API units in the Tulare Formation to lower overall API units in the San Joaquin Formation. These criteria can be observed in the geophysical log in S1 Fig. In the eastern part of the Coles Levee Oil Fields, the upper San Joaquin Formation becomes very sand rich and the formations are nearly impossible to distinguish on the logs. Geophysical logs and well history data used for the study are available online on the CalGEM Well Finder website

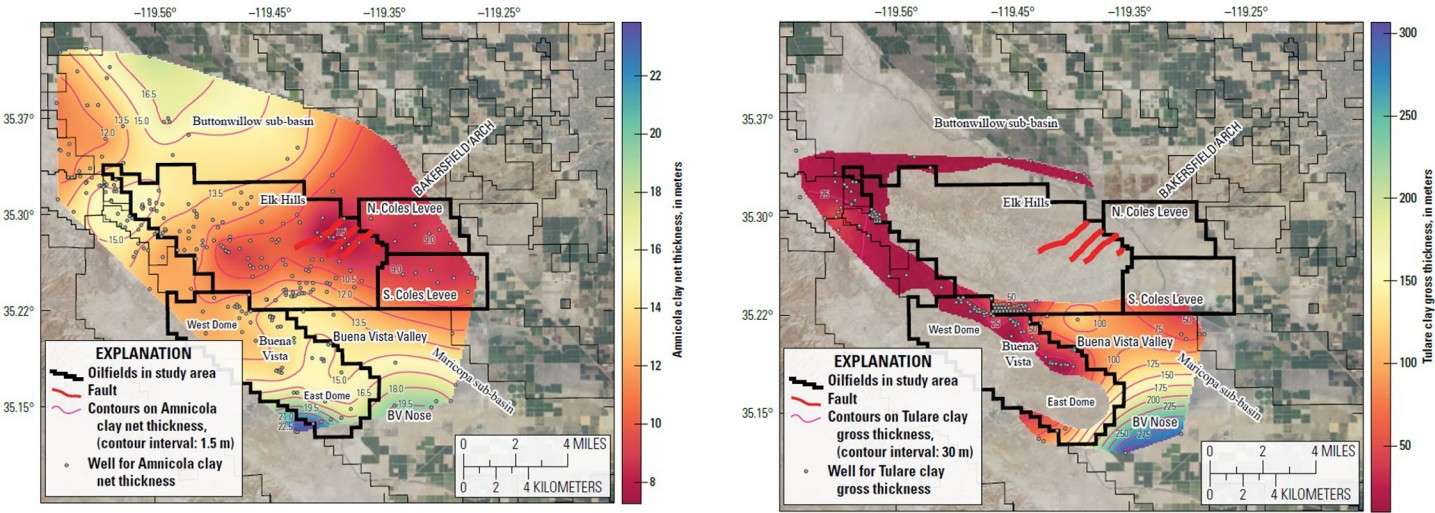

**Fig 6. Thickness isochore maps of major clay layers in the Tulare Formation.** Heavy black outlines mark oilfield administrative boundaries. a) Net clay isochore map of the Amnicola clay (gross thickness minus sand interbeds). Wells shown are those used to generate the contours. Contour interval = 1.5 m. b) Gross interval isochore map of the Tulare clay. Wells shown are those used to generate the contours. Contour interval = 25 m. Air photo base from https://basemap.nationalmap.gov/arcgis/rest/services/USGSImageryOnly/MapServer/.

(https://www.conservation.ca.gov/calgem/Pages/WellFinder.aspx, accessed 5/15/2020). The mapped extent for the base of the Tulare Formation, the Amnicola clay, and the Tulare clay differ because of differences in the geographic extent of the data available to map these surfaces with confidence. Input data for all maps are available in Stephens et al. [23].

Water-table elevation maps were created to visualize potential groundwater flow directions between the oil field areas and the adjacent groundwater basin (Fig 7). The maps were constructed using two data sources: 1) water-surface elevations (WSE) in groundwater wells from the California Department of Water Resources (DWR) online database for periodic groundwater level measurements [53], California State Water Resources Control Board [54], and, for one well, Wood Environment and Infrastructure Solutions, Inc [55]; 2) elevation of the top of the water table as interpreted from geophysical (density-neutron) well log scans collected when oil/gas production or injection wells were drilled and archived in the California Geologic Energy Management online database [51]. The two data sources rarely cover the same areas, because groundwater level data generally occur adjacent to oil fields and the geophysical logs occur within oil fields. Both datasets are discussed further below. Because the data coverage is discrete in space and time in the study area, we used a Gaussian process described below to interpolate the data to create continuous water table maps for selected years. The WSE from CA DWR [53] are collected by CA DWR and other agencies from wells within groundwater basins throughout the state. WSE are typically measured twice per year to capture the high and low periods; however, data are collected more frequently from some sites [53]. Of these data we used only a subset near Elk Hills Oil Field and after 1975. There are 276 unique sites in the study area, with 11,391 water level measurements from 1975 to 2019.

https://www.arcgis.com/home/item.html?id=1b243539f4514b6ba35e7d995890db1d. (September 30, 2021).

The top of the water table in geophysical logs was found in one of two ways:

1. Observing the lowest depth of crossover between neutron and density porosity logs—interpreted to represent air sands, or unsaturated zone (S2 Fig).

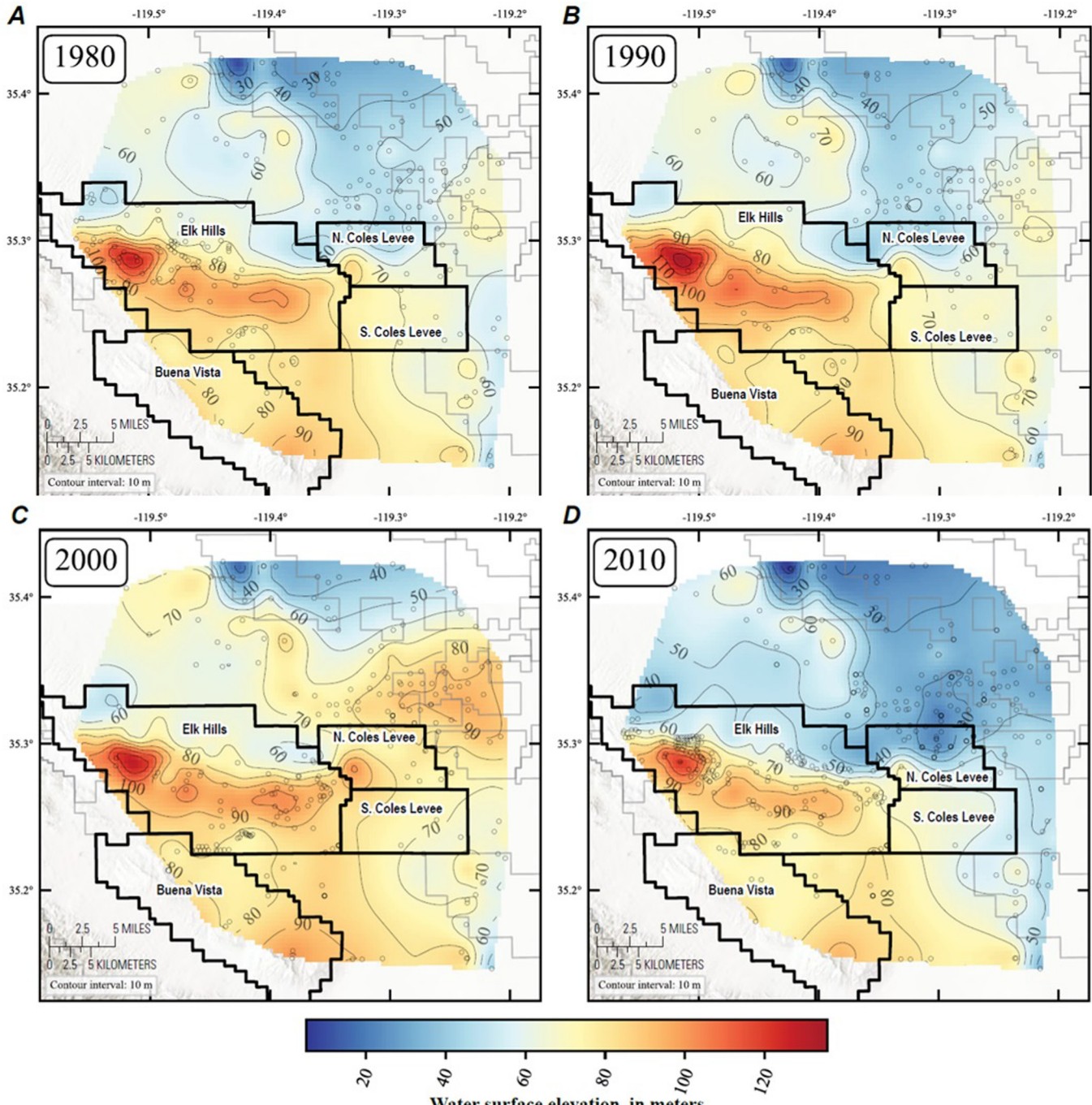

**Fig 7.** Groundwater surface elevation maps for the years a) 1980, b) 1990, c) 2000 and d) 2010. The maps were constructed from water surface elevations in groundwater wells outside the oilfields and from density-neutron log data showing the base of the vadose zone at the time of drilling in wells within the oilfields. Because the data coverage is discrete in space and time in the study area, a Gaussian process was used to interpolate the data to create continuous water table maps for selected years and is described in the supplementary material. Contour interval = 10 m. Base map from Esri. "World Hillshade" [basemap]. Scale Not Given. "World Hillshade". July 9, 2015.

2. Where porosity logs are not available on some of the older logs, resistivity curves were used to identify base of the vadose zone in the western part of the study area. In this area, resistivities of sands in the vadose zone are considerably higher than those in the brackish water

sands in the upper aquifer. This pattern allows for easier identification of the water table than farther east where groundwater is fresher and has higher resistivities more similar to those of unsaturated sands. In previous work in the study area, the top of the water table was picked where resistivity dropped from greater than 10 ohm-meters (ohm-m) to less than 5 ohm-m [42].

Water table depths were estimated from 351 geophysical logs in the Elk Hills Oil Field area in wells drilled from 1976 to 2014. The depths were converted to elevations using the elevations of the kelly bushing or derrick floor and subtracting 3 m (10 ft is the most common height of the rig floor above ground level) to estimate the ground surface. The WSE model described below was used to produce a probability distribution for the WSE of an array of points in the studied oil fields and the adjacent groundwater basin. The WSE were modeled using a Gaussian process with a non-stationary mean [56]. The equations, solution approaches, and validation approaches used in the WSE model are described in the supplemental material. A flow chart describing the steps taken to create the WSE model is shown in S3A and S3B Fig. The model code, input, and output data are available from Stephens et al. [23].

## Groundwater salinity

Groundwater salinity in the study area was determined using two methods: 1) historical data from lab analysis of water samples collected from oil and groundwater wells completed in the Tulare Formation within the oil fields and 2) geophysical log analysis. Water sample data from 16 wells in the Tulare Formation in the study area were used in the study. The analyses were found on the CalGEM FTP site (ftp://ftp.consrv.ca.gov/pub/oil/chemical_analysis; accessed 2/28/2020), in aquifer exemption applications submitted to CalGEM by California Resources Corporation [42, 52] and, in the case of well MW 58–30 (30S25E30Q01) from Wood Environment and Infrastructure Solutions [55]. The water sample data and associated metadata for this study are available in Stephens et al. [23]. Sample collection dates ranged from February 27, 1965, to June 28, 2019, and some of these wells were sampled numerous times during the 1990s. Most of the wells with multiple samples are water-source wells used to supply water for petroleum operations in the Elk Hills Oil Field. While direct measurement of salinity using lab analysis of water samples is generally the preferred method for determining salinity, sample data have some drawbacks. The sampled wells are not uniformly distributed across the study area and the samples represent a mixture of water from within the perforated interval—therefore, it is not possible to determine the percentage of water contributed to the sample by each sand within the perforated interval. Since the sample comes from a single perforated interval within the aquifer, it is not possible to determine salinity gradients with depth in the aquifer. Water sample data from groundwater wells outside of the oil fields were not included in the study because these samples are generally fresh and comparatively shallow compared to the depths of the borehole geophysical logs and therefore did not contribute to defining the distribution of salinity in the study area.

Geophysical logs are numerous within the study area—especially in the western part where groundwater quality is relatively poor and few water wells are present. The Coles Levee Oil Field area is not as well represented because many of the wells were not logged above 450 m (1,500 ft) and because most of the logs are in wells drilled before 1975, prior to the use of density-neutron porosity logs. Using resistivity and density-neutron logs, salinity values (as sodium chloride (NaCl)) can be calculated for sand layers in a vertical profile of the aquifer where logs cover the Tulare Formation.

Use of geophysical logs to determine water salinity is common in the petroleum industry. The resistivity of the water ($R_w$) dispersed through the oil reservoir is necessary in order to

determine the oil saturation. The Archie [57] equation or one of its many variants is typically used. Archie [57] related the *in situ* electrical resistivity of a 100% water saturated sedimentary rock ($R_o$) to its porosity ($\phi$), non-dimensional factors related to matrix properties ($m,a$) and the formation water resistivity ($R_w$) as shown in Eq 1.

$$R_w = R_o * (\phi^m/a) \qquad (1)$$

$R_o$ is the resistivity of a clean (low clay content), wet (oil and gas-free) sand and is obtained from the deep resistivity curve [57]. Sands were identified by evaluating nearby core, mudlog, and drillers log data. These data allowed us to ascertain that the analyzed sands did not contain oil or gas. Generally, the porosity value is obtained from porosity logs such as density, neutron, sonic or nuclear magnetic resonance (NMR) logs or from core analysis. Sonic porosities were not used in our study because the sonic porosity values are generally too high in unconsolidated formations and must be corrected using a compaction factor [58]. Since the compaction factor is unknown in this area, we prefer the use of density-neutron or NMR logs for porosity values.

Many of the aquifer sediments are poorly sorted and some sands contain large amounts of clay and silt. Therefore, the porosity values used for the log analysis calculations of salinity averaged the porosity for each depth pick by weighting the density porosity values twice as much as the neutron values. Neutron logs calculate porosity by measuring the hydrogen concentration in the rocks encountered in the borehole. Most clays contain hydroxide groups in their crystal lattice, so the neutron curve overestimates porosity in clay-rich zones and generates a porosity value in clays that is much higher than that generated by the density log. The largest difference measured between the density and neutron curves within an aquifer zone was considered to be 100% clay. The difference between the density and neutron curves for each analyzed interval was divided by the 100% clay separation value to estimate percent clay content [58]. The $R_w$ values were not used for analyzed intervals containing over 25% clay. Domenico and Schwartz (Table 2.1) [59] showed that sand deposits can have porosity values that range from 24–53% with the higher porosities occurring in finer grained sands. Because detrital diatomite is common in the study area, and the low density of diatomite can cause the density curve to record extremely high porosity values, we excluded intervals with density porosity greater than 45%.

The exponent *m* from Eq 1 is the cementation factor (unitless) and is related to the amount of cementation in clastic rocks [57]. It typically ranges from 1.3 to 3 [60]. The value *a* (unitless) is related to the length the electrical current must travel through the rock and is called the tortuosity factor [57]. The value for *a* is typically between 0.5 and 1.5. Many studies, including aquifer exemption applications in the study area from California Resources Corporation [42, 52], rely on values for *a* and *m* obtained from other areas such as the Texas-Louisiana Gulf Coast (Humble equation parameters $a = 0.62$, $m = 2.15$; [61]). However, for this study, core analyses to determine the formation factor (F) (where $F = a/\phi^m$) were available from five intervals in the Tulare Formation in a water disposal well (API 0402959052) in the Buena Vista Oil Field in the CalGEM online files. The value for *a* was held constant at 1 while *m* was allowed to vary. The calculated values for *m* ranged from 1.62 to 1.87 with a mean value of 1.79 (standard deviation of 0.1) and a median value of 1.8. For our salinity calculations we used $a = 1$ and $m = 1.8$ based on the core analysis results.

After obtaining a value for $R_w$ and determining the temperature of the interval of interest (FT) by a linear interpolation between bottom hole temperature and a mean annual air temperature of 18°C (65°F) [62], the formation water salinity was estimated from the Bateman and Konen [63] Eq (2) relating fluid resistivity at 75°F (($R_{w75}$) Eq (3) to NaCl equivalent TDS

concentration (note that the Bateman and Konen equation [63] uses temperatures in degrees Fahrenheit).

$$\text{TDS mg/L NaCl @ } 75°F = 10^{((\log R_{w75} - 0.0123)/0.955)} \qquad (2)$$

$$R_{w75} = R_w \text{ x } (FT + 6.77)/81.77 \qquad (3)$$

Calculated salinities are shown in Stephens et al. [23], and a flow chart showing the steps taken to calculate salinity is shown in S3C Fig.

## Results

### Mapping lithologic and water-table surfaces

Fig 5 shows the depth from the land surface to the base of the Tulare Formation and represents the thickness of the formation and overlying alluvium. The Tulare Formation thins to approximately 305 m (1,000 ft) over the crests of the Elk Hills and Buena Vista anticlines and thickens to as much as 1,220 m (4,000 ft) to the north and southeast into the Buttonwillow and Maricopa sub-basins respectively.

The net clay map of the Amnicola clay is shown in Fig 6A. The map was constructed by differencing the top and base depths of the clay and then subtracting sand interval thicknesses within the clay noted on the SP log resulting in the net clay thickness. Because the clay contains numerous sand intervals (especially in the east), mapping the net clay thickness provides a better understanding of the ability of the Amnicola clay to act as a regional confining layer. The clay is present throughout the study area, with the possible exception of the eastern portions of North and South Coles Levee Oil Fields. The net clay thickness ranges from 15 to 24 m (50 to 80 ft) in western Elk Hills Oil Field to less than 6 m (20 ft) in the eastern part of the Coles Levee Oil Fields where it is highly variable due to the presence of numerous sand layers within the clay. During the Plio-Pleistocene, this area was, and remains, a major depocenter for coarse clastic sediments carried from the Sierra Nevada by the Kern River. As a result, the Amnicola clay interval contains a high percentage of sand and clay correlations are difficult. The clay appears thickest (greater than 30 m (100 ft)) in the southern Buena Vista Oil Field. Here the clay contains several high resistivity intervals with no corresponding SP effects to indicate permeability—these intervals may represent low permeability, fresh-water carbonates; however, core data are not available to support this interpretation.

The Tulare clay gross thickness map is shown in Fig 6B. Sand content is difficult to obtain for this layer because the upper contact occurs near the surface and the uppermost part of the clay was not logged in many wells. Because the number and thickness of sand layers in the upper part is unknown, net clay was not mapped for the Tulare clay. The Tulare clay is only present around the southern, western and, to a lesser extent, northern margins of the Elk Hills anticline (Fig 6B). Its thickness is much more variable than that of the Amnicola clay. The Tulare clay is over 305 m (1,000 ft) thick in the Maricopa sub-basin in the southeastern part of the study area but is absent over the crest of the Elk Hills and Buena Vista anticlines and does not appear to be present in the Coles Levee Oil Field area (although shallow well log data in this area are limited). It appears to be present, but much thinner, northeast of the Elk Hills Oil Field; but, in this area, the sand content is high and correlations are difficult.

The relationship between the Tulare clay and the Corcoran Clay (also known as the E-clay of Croft [45]) is uncertain. The disruption in lacustrine sedimentation caused by the Elk Hills uplift in the west and the Kern River alluvial fan in the east makes correlation tenuous between the more well-defined Corcoran Clay in the Buttonwillow sub-basin (Tulare Lake area) to the north and the Maricopa sub-basin (Buena Vista Lake area) to the south. Croft (Plate 2) [45]

noted the presence of an ash bed below a 6 to 9 m (20 to 30 ft) thick blue clay in core at a depth of approximately 186 m (613 ft) in well 31S25E27F01 in the Buena Vista Lakebed, and Croft believes this may represent the E-clay because ash beds have also been noted below the E-Clay to the north in the Buttonwillow area. The Tulare clay in 31S25E27F01 mapped in this study is 61 m (200 ft) thick with its top at a depth of 217 m (711 ft) and is separated from the E-Clay by a 15 m (50 ft) sandy interval. In Croft's [45] well 31S24E22G (well API 0402938056 in this study), 15 km (9.5 miles) to the west, the E-clay is 11 m (35 ft) thick and lies 152 m (500 ft) below the surface. The underlying Tulare clay is 175 m (575 ft) thick and separated from the E-clay by a 55 m (180 ft) sandy interval. A cross section which includes the two wells used by Croft [45] is shown in S4 Fig to illustrate the relationship between the Tulare clay and Corcoran Clay. The Corcoran Clay was mapped in the study area by Croft [45] and Faunt [39] and is not mapped as a separate clay interval in this study.

The combination of water-table elevations estimated from historical geophysical logs in the oil fields with water-level measurements in water wells outside the oil fields indicate that lateral groundwater potential flow gradients are from the oil field areas toward the groundwater basin in the valley (Fig 7). Fig 7 shows the water surface elevation model results for four different years: 1980, 1990, 2000 and 2010. The water table is consistently highest in the Elk Hills and Buena Vista Oil Fields, reflecting that the base of the Tulare Formation occurs at higher elevations in these areas, creating a topographic gradient. Notably, the Tulare aquifer consists mainly of a thick vadose zone in the central parts of these oil fields; however, as will be discussed in a later section, saline produced water is being injected in increasing volumes into the vadose zone along the crest of the Elk Hills Oil Field, providing an artificial source of aquifer recharge along the crest of the anticline. The northeast part of the study area has the lowest groundwater elevation during each time period. An increase in the elevation of the water table in the eastern part of the study area caused by recharge in the Kern Water Bank can be seen in the map from 2000 (Fig 7C). However, the ensuing drought shows that the water table had fallen in this area by 2010 (Fig 7D) as water was removed from the aquifer near the Kern Water Bank.

Some caution should be used in the interpretation as the WSE measurements from both the upper and lower Tulare Formation are combined in these maps. While most of the measurements are from the upper Tulare Formation, measurements from the lower Tulare Formation are common along the crest of the Elk Hills and Buena Vista Hills structure where the Tulare Formation is largely unsaturated. In addition, completion depths are not always available for some of the water wells used in the maps so it is unknown if the WSE measurements refer to water levels from the upper or lower Tulare Formation. The Gaussian process model used to create the WSE maps provides a probability distribution for WSE at each location; therefore, we can quantify the uncertainty associated with each WSE prediction. At each location, the model provides a mean and variance, taking the square root of the variance gives the standard deviation ($\sigma$), which is mapped in S5 Fig. The $\sigma$ maps should be considered with the WSE maps (Fig 7) to understand which zones have higher or lower uncertainty.

## Groundwater salinity

Salinity versus depth plots of log-calculated salinity were compared to TDS values of laboratory analyses of water samples from 16 wells completed in the Tulare Formation (Fig 8). Because vertical salinity gradients are steeper in areas where the elevation of the base of the Tulare Formation is high and flatter where it is low, individual log-calculated TDS versus depth plots were prepared for wells in areas where the base of the Tulare Formation was at an elevation of a) above -152 m (-500 ft) (Fig 8A), b) between -152 and -305 m (-500 and -1,000 ft) (Fig 8B), c)

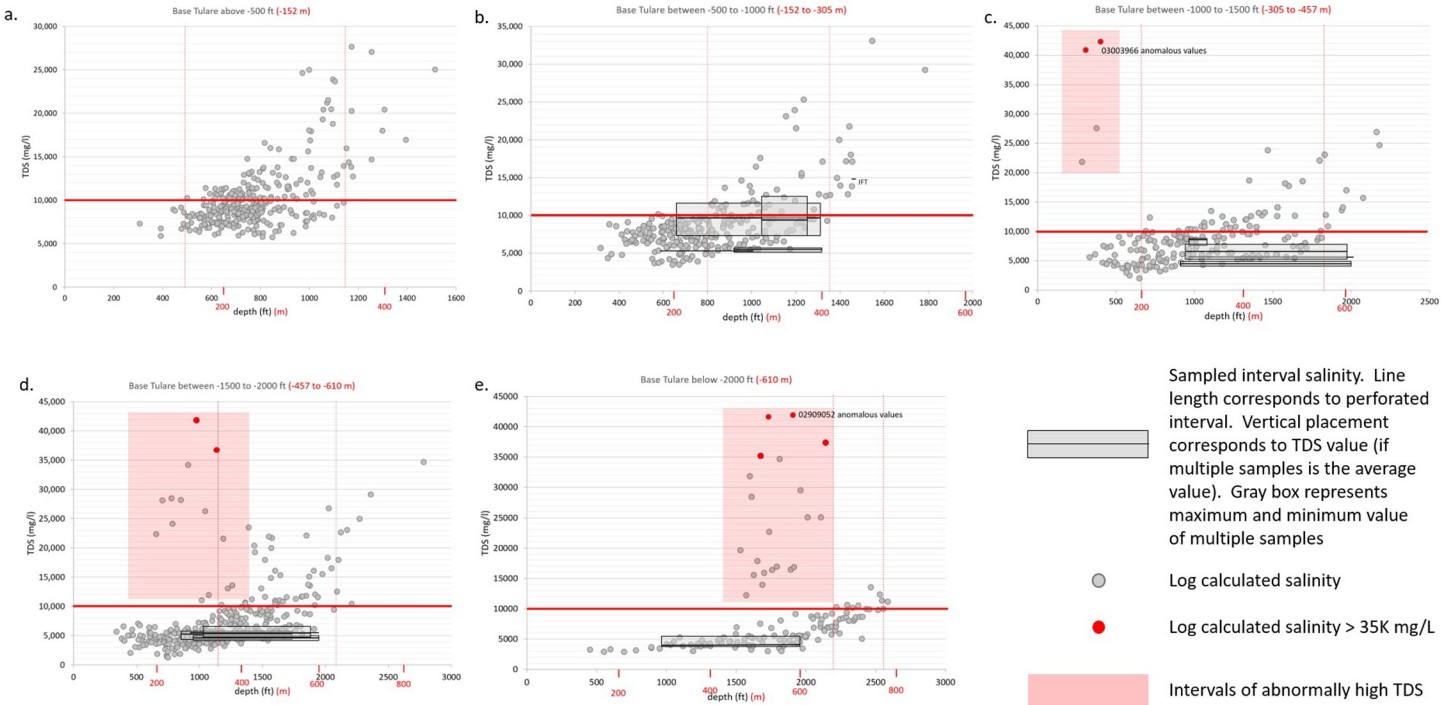

**Fig 8. Log-calculated total dissolved solids (TDS) versus depth plots for wells within different elevation ranges for the base of the Tulare Formation.** Log calculated TDS shown by the gray dots. Wells with lab sample analyses are shown as black horizontal lines between the upper and lower perforated interval depths. The vertical position of the line corresponds to the salinity value of the sample. For wells with multiple samples, the highest and lowest salinity values are shown by a gray box and the horizontal line is placed at the average salinity value. Red boxes note log calculated salinity values that are abnormally high and are sometimes associated with produced water disposal wells. The red horizontal line on each graph marks the 10,000 mg/L TDS value. Red vertical dashed lines mark the upper and lower depths at which groundwater salinity exceeds 10,000 mg/L within each depth range. Points with calculated salinities greater than sea water are shown in red. Values in feet in black, values in meters in red.

between -305 m and -457 m (-1,000 and -1,500 ft) (Fig 8C), d) between -457 and -610 m (-1,500 and -2,000 ft) (Fig 8D), and e) deeper than -610 m (-2,000 ft) (Fig 8E). Lab analyses for water samples from wells were plotted on the log-calculated salinity graphs that corresponded to the elevation for the base of the Tulare Formation in that well. Red boxes indicate salinity values that are anomalously high in comparison with predominant vertical salinity gradients. Where wells with higher than normal calculated salinity values are in close proximity to water disposal injection wells or ponds, they are likely to have been affected by disposal injection or percolation of high TDS produced waters from the Stevens sands (Monterey Formation) and San Joaquin and Etchegoin Formations oil reservoirs.

The log calculated salinities indicate a range of depth values for Base 10K within each elevation interval as shown by the dotted vertical red lines (Fig 8). A comparison of groundwater sample analysis to the log calculations shows generally good agreement; however, some salinities calculated from geophysical logs are higher than the salinity of some of the lab-analyzed samples—especially for the deeper parts of the perforated intervals. The higher calculated salinities from the geophysical logs suggests that the depth to Base 10K may be slightly deeper than that indicated by the log analysis. However, it could also mean that the upper part of the perforated interval contributed more to the samples analyzed in the lab. Eight calculated salinity values (out of 1,568 total) are in excess of 35,000 mg/L which also suggests that the calculated salinities may be somewhat high since injected disposal water from the Stevens sand, Etchegoin Formation and lower San Joaquin Formation oil-producing reservoirs at Elk Hills

Oil Field have TDS values of 27,000 and 28,000 mg/L, respectively [42]. However, four of the eight anomalously high calculated salinity values come from a single WD well in Buena Vista Hills (0402959052, Fig 8E). This log exhibits low resistivity values throughout and was drilled with low resistivity mud. An additional two values come from very shallow depths (90 to 120 m (300 to 400 ft)) in perched aquifers within the vadose zone in well 0403003966 (Fig 8C). The source of these high salinity waters at such shallow depths in the aquifer far from major WD injection is unknown but the high values may represent concentration by evaporation of water at the surface that subsequently percolated into the vadose zone. No disposal ponds are apparent on recent aerial photos in this location.

Using log calculations from 152 wells throughout the study area, the depth at which water salinity reached 10,000 mg/L TDS was calculated. The resulting values were contoured to provide a map showing measured depth to 10,000 mg/L TDS or Base 10K (Fig 9). The transition to water with TDS greater than 10,000 mg/L (base 10K) typically occurs between the Amnicola clay and the base of the Tulare Formation. As a result, the depth to base 10K varies according

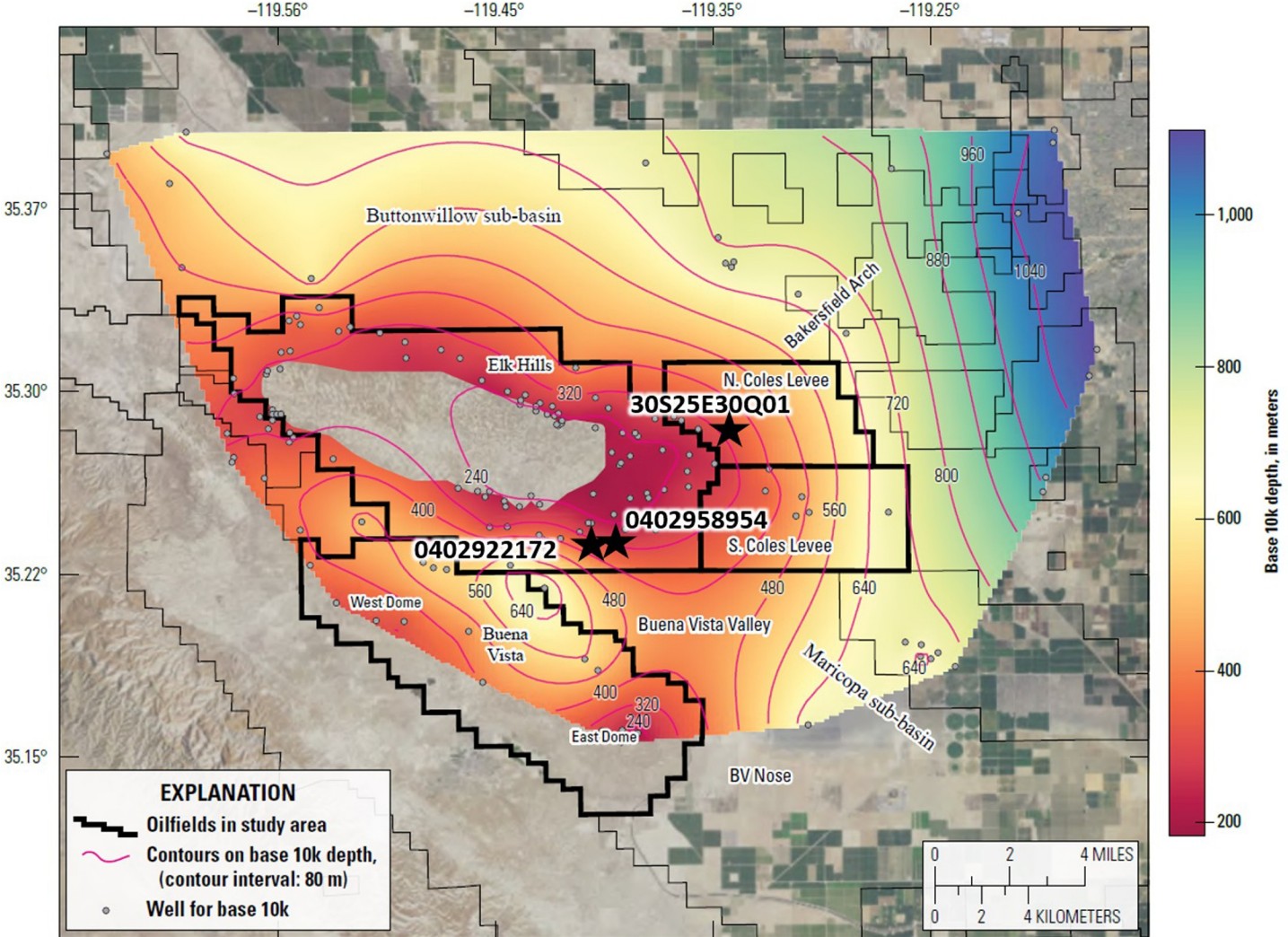

**Fig 9. Map showing depth to log-calculated salinity of 10,000 mg/L total dissolved solids (Base 10K) in the Tulare aquifer.** Wells shown are those used to generate the contours. Empty area over Elk Hills Oil Field contains no groundwater with less than 10,000 mg/L TDS. Black stars show location of wells in Table 2. Air photo base from https://basemap.nationalmap.gov/arcgis/rest/services/USGSImageryOnly/MapServer/.

to the elevation of the base of the Tulare Formation—it is shallower in areas where the base of the Tulare Formation lies at a higher elevation along the crests of the anticlines which host the oil fields and deeper within the adjacent synclines such as the Buena Vista Valley (Fig 9). The depth to Base 10K is shallowest (about 245 m (800 ft)) across the crest of the Elk Hills anticline and on the east dome at Buena Vista Oil Field. It becomes progressively deeper to the east— about 730 m (2,400 ft) deep in eastern Coles Levee Oil Fields and 1,100 m (3,600 ft) deep farther east on the Bakersfield Arch. The greater depth to Base 10K in the eastern part of the study area is due to the presence of the Kern River, which flows through this area and recharges the underlying Tulare aquifer. No large source of fresh-water recharge is present in the western part of the study area.

The Tulare aquifer is nearly entirely unsaturated along the crest of the Elk Hills anticline (as shown by the empty areas on the map along the crest of the anticline in Fig 9). Here there is little or no groundwater with TDS less than 10,000 mg/L.

Three groundwater samples (Table 2) have average salinities approaching 10,000 mg/L— two from oil wells recompleted in the Tulare Formation as water source wells and one from a monitoring well in the western North Coles Levee Oil Field. Comparing the completion depths of these wells to the depth to Base 10K map (Fig 9) provides an additional check on the accuracy of the map. In each case, the mapped depth to Base 10K from log analysis at the location of the wells falls within the completion interval of the three wells.

By subtracting the water table depth from the Base 10K depth, the thickness of the aquifer containing water with less than 10,000 mg/L TDS may be determined. This thickness is shown on the map in Fig 10. The thickness of groundwater with less than 10,000 mg/L TDS is zero in the empty areas on the map along the crest of the Elk Hills anticline and the thickness increases on the flanks of the anticline into the adjacent syncline in the Buena Vista Valley. In the eastern part of the Coles Levee Oil Fields, the thickness of the aquifer with water less than 10,000 mg/L TDS increases to 670 m (2,200 ft).

The vertical distribution of water with less than 10,000 mg/L TDS across the Elk Hills anticline is illustrated in the cross section in Fig 11. The vadose zone (gray shading) is up to 245 m (800 ft) thick over the crest of the Elk Hills anticline and the base of the vadose zone (red horizon) is at nearly the same depth as the base of the Tulare Formation (black horizon). The depth to Base 10K is shown by the blue horizon on the cross section and occurs within the lower Tulare Formation between the Amnicola clay and the base of the Tulare Formation. Groundwater with less than 10,000 mg/L TDS is shaded blue between the Base 10K depth and the base of the vadose zone. The water with less than 10,000 mg/L TDS forms wedges that are thickest on the northern and southern flanks of the Elk Hills structure and taper to zero near the crest.

## Effects of oil field activities

In and near water disposal areas, the replacement of brackish Tulare Formation water by higher salinity injectate has resulted in the development of anomalous deep resistivity profiles

**Table 2. Completion depths for sampled wells with salinity values near 10,000 mg/L TDS compared to mapped depth to water with 10,000 mg/L TDS at the wells' locations in the study area as a check on map accuracy.**

| API or water well number | field | average TDS (mg/L) | perforated interval (ft) | perforated interval (m) | Mapped depth to base 10K (ft) | Mapped depth to base 10K (m) |
|---|---|---|---|---|---|---|
| 0402929172 | Elk Hills | 9,541 | 1040-1275 | 317-389 | 1,050 | 320 |
| 0402958954 | Elk Hills | 9,558 | 915-1315 | 279-401 | 950 | 290 |
| 30S25E30Q01 | No. Coles Levee | 8,663 | 934-1071 | 285-326 | 1,100 | 335 |

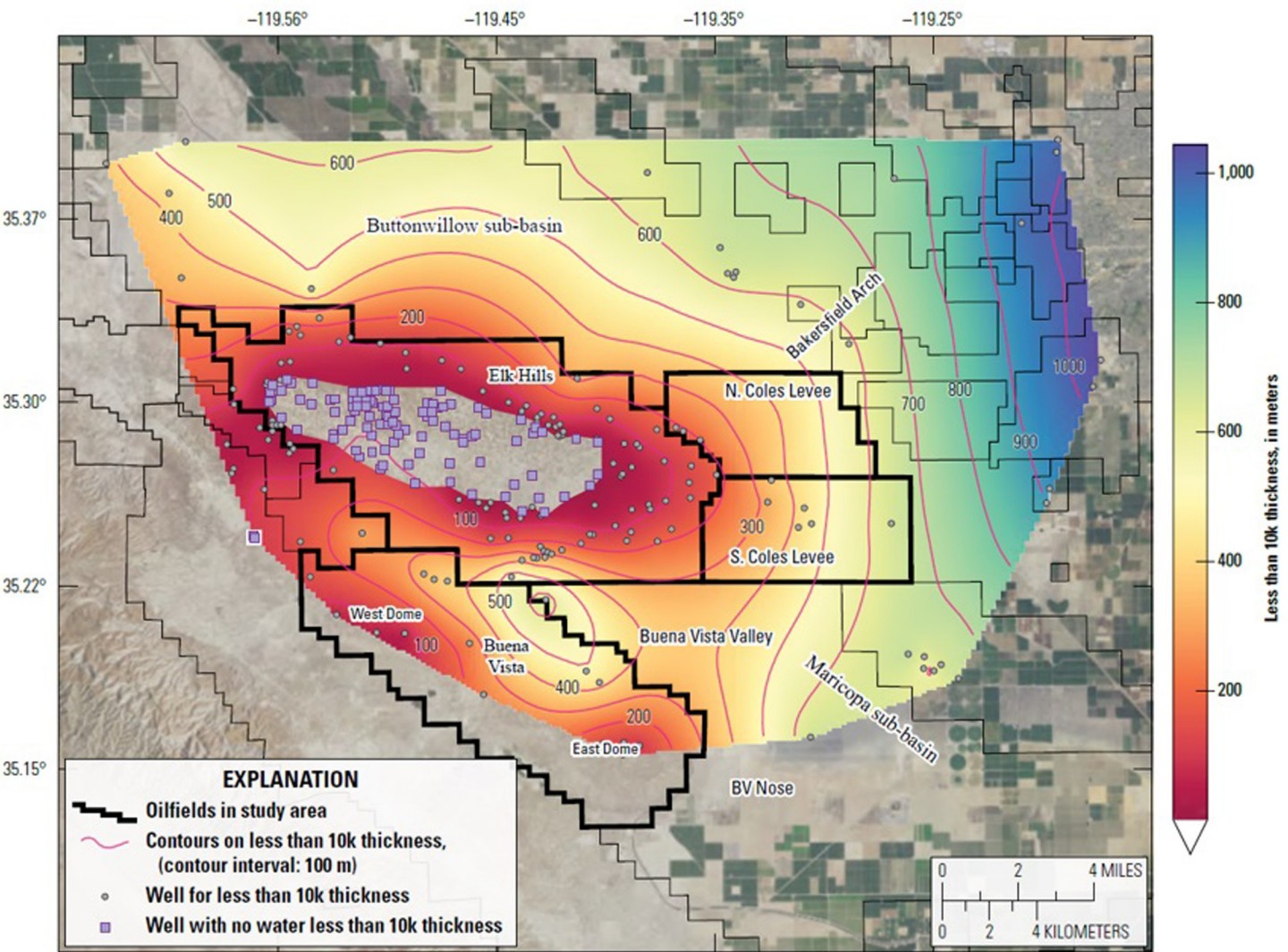

**Fig 10. Map showing the thickness of the Tulare aquifer interval containing water with Total Dissolved Solids (TDS) less than 10,000 mg/L.** Empty area in Elk Hills Oil Field contains no groundwater with less than 10,000 mg/L TDS. Wells shown are those used to generate the contours. Heavy black outlines mark oilfield administrative boundaries. Air photo base from https://basemap.nationalmap.gov/arcgis/rest/services/USGSImageryOnly/MapServer/.

on electrical logs in wells drilled after the initiation of disposal (Fig 4C). Log-calculated salinities in wells logged prior to the initiation of large scale WD activity display a relatively smooth increase with depth whereas the wells drilled after the initiation of water disposal in these areas display a much more variable salinity profile with depth. Logs in wells affected by injection of saline produced water show sands that exhibit abnormally low deep resistivity values for their depth—sometimes less than half the deep resistivity of the older, unaffected wells at the same depth. In some cases, only the lower part of an individual sand contains low resistivity saline water while the waters in the upper part remain brackish and exhibit higher resistivity values, suggesting that the saline injected water and brackish formation waters are segregated by density and mixing has not had time to occur in some sands (S6 Fig). Similar behavior has been noted in experiments by Shincariol and Schwartz [64]. In their study they injected water of varying salinity (1000, 2000, 10,000 and 100,000 mg/L NaCl) into an experimental tank filled

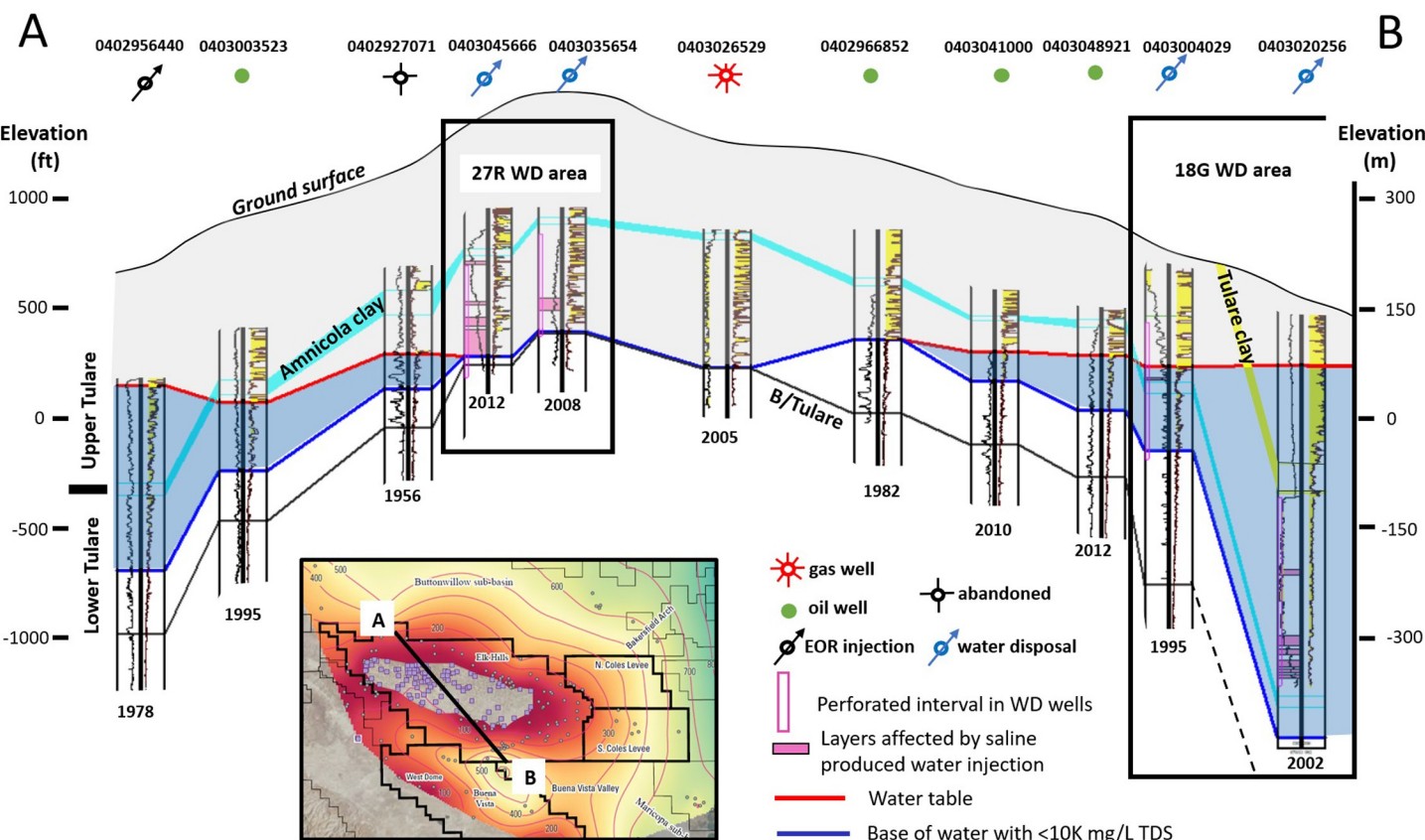

**Fig 11. Northwest cross section A-B across the Elk Hills Oil Field.** Location is shown in inset map at bottom of figure. Well API numbers used to identify oil and gas wells in the United States are shown at the top of each well and year drilled and logged is shown below the log. Gray shaded area marks the vadose zone. Blue shaded area denotes groundwater with less than 10,000 mg/L TDS (Base 10K). Well use abbreviations: WD = water disposal, EOR = enhanced oil recovery. Left track of each log is the spontaneous potential (SP) curve shaded yellow for sand layers. Right track contains the deep and shallow resistivity curves. Deep resistivity above three ohm-m is shaded yellow. Given the temperature gradient in the study area, sands with deep resistivity values above three ohm-m usually contain water with less than 10,000 mg/L TDS. Air photo base on inset map from https://basemap.nationalmap.gov/arcgis/rest/services/USGSImageryOnly/MapServer/.

with porous media and fresh water to observe its behavior. The denser, more saline plumes sank to the bottom of the tank after 13 hours and, at salinities of 25,000 mg/L and greater, flowed into the tank by gravity at a rate greater than the pump rate. More pertinent to our study is their observation that "when layered media were placed in the experiment tank, dense plumes mounded above the boundary with a low conductivity unit and behaved in a manner not unlike that of a dense, non-aqueous phase liquid" [64]. We note similar behavior by injected saltwater into the brackish water aquifer in our study—the injected saline water appears to form a plume that flows along the base of the sand bodies.

Additionally, density-neutron curves in wells affected by disposal of produced waters display thin intervals (approximately 1 m (3 ft)) of cross-over (S6 Fig). McMahon et al. [65] noted the presence of both thermogenic gases and dissolved organic carbon in produced water injectate in the Lost Hills Oil Field, located 23 km (15 mi) northwest of the study area. Assuming the injectate in the study area also contains these components, the observed density neutron cross over may be caused by either thermogenic methane in the injectate or bacterial metabolism of the dissolved organic matter in the injectate resulting in the generation of biogenic methane *in situ*.

Fig 12A is an example graph of log-derived salinity calculations with depth from a well logged near the start of water disposal injection in 1982 (API 0402967266) in the 18G water

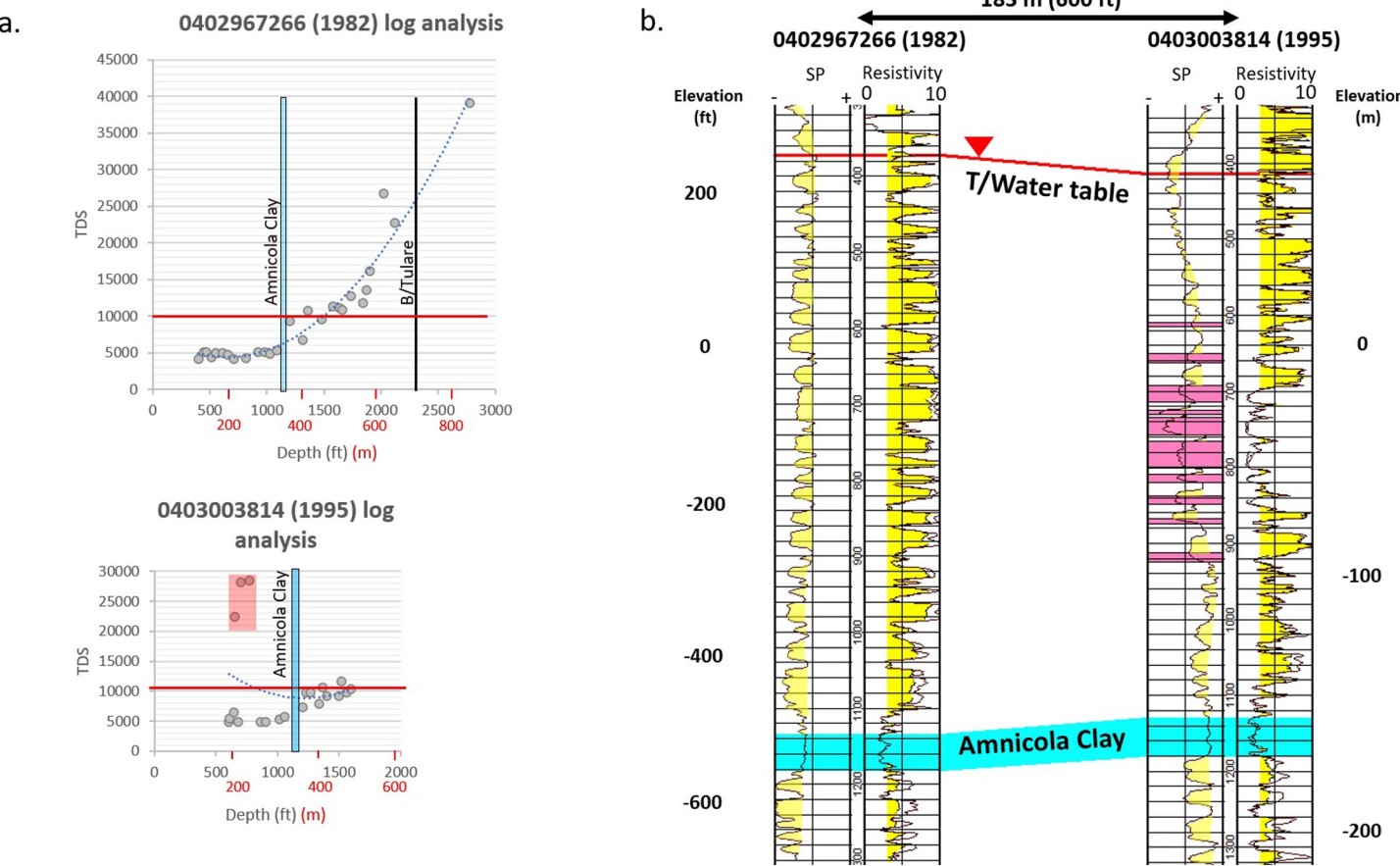

**Fig 12. A comparison of two wells located 610 m (2000 feet) south of the initial disposal operations that commenced in 1981 in the Elk Hills Oil Field 18G WD area (location shown in Fig 13B).** Depths between the wells do not align exactly due to differences in the elevation of the rig floor from which the depths were measured. Well API 0402967266 was drilled and logged in 1982 near the beginning of WD injection. Well API 0402967266 was drilled and logged 183 m (600 ft) east in 1995—thirteen years later. a) The TDS versus depth graphs for each well with abnormally high calculated salinity values highlighted in red for the 1995 well. b) The log response (SP in the left track and shallow and deep resistivity in the right track) of the wells between 91 and 400 m (300 and 1,300 ft). Sand intervals are highlighted in yellow on the SP curve and resistivities above three ohms are highlighted in yellow on the deep resistivity curve. Low resistivity anomalies in the 1995 well are highlighted in pink.

disposal area in the southern Elk Hills Oil Field. This well is compared to a nearby well drilled in 1995 (API 0403003814)—approximately 13 years after the start of disposal operations (locations shown in Fig 13B). The wells are approximately 600 m (2,000 ft) south (downdip) of the initial disposal operations and are 180 m (600 ft) apart. Deep resistivity values in sands between 180 and 275 m (600 and 900 ft) are approximately 8 to 9 ohm-m in the 1982 well (Fig 12B). In the 1995 well, the lower parts of some of these sands have resistivity values between 1 and 2 ohm-m. Intervals of abnormally high salinity (greater than 20,000 mg/L TDS) on the graph can be observed in the deep resistivity curve in the 1995 well between 150 and 300 m (500 and 1,000 ft) (pink shaded depth intervals in Fig 12B) whereas the older 1982 well shows salinities ranging from 4,000 to 5,000 mg/L on the graph within the same depth interval. Sands with log-calculated anomalously high salinity waters from this and other locations are shown in red boxes in the TDS versus depth plots in Fig 8.

By far the largest volume of disposal by injection into the Tulare Formation has been in the 18G area on the south flank of the Elk Hills Oil Field (Fig 4A). Since 1980, over 121 million m³ (762 million barrels) of produced water have been injected into the upper Tulare Formation in this area. Continued post-disposal drilling and logging of wells in various

years in and near this area provides an opportunity to observe the movement of the injected fluids over time by noting the presence or absence of resistivity anomalies in the logs from the new wells. The Tulare Formation dips southward in the area with dips ranging from 4-9˚ in the northern part of the area and exceeding 27˚ in the south [30]. The Tulare clay crops out in this area [30] and initially (1980's; Fig 13A), disposal occurred north of the outcrop area (shown by the black dash-dot line) where the Tulare clay was not present to act as a confining layer.

By the mid-1990s (approximately 13 years after the initiation of disposal), disposal of produced water in the area had resulted in five surface expressions (Fig 13B). Resistivity anomalies in geophysical logs show that injected saline water had traveled approximately 425 m (1,400 ft) downdip to the southeast (Fig 13B) as indicated by the distance between the nearest WD well and the 1995 well shown in Fig 12B. Groundwater withdrawal from a series of water source wells (blue triangles) completed below the Tulare clay south of the 1980's disposal field likely aided the southward migration of the plume. These water source wells (approximately 850 m (2,800 ft) south of the 1980's WD wells) were sampled periodically throughout the 1990's but did not show increases in salinity at that time. No wells were drilled and logged through the Tulare Formation within 600 m (2,000 ft) to the north (updip) in the 1990s.

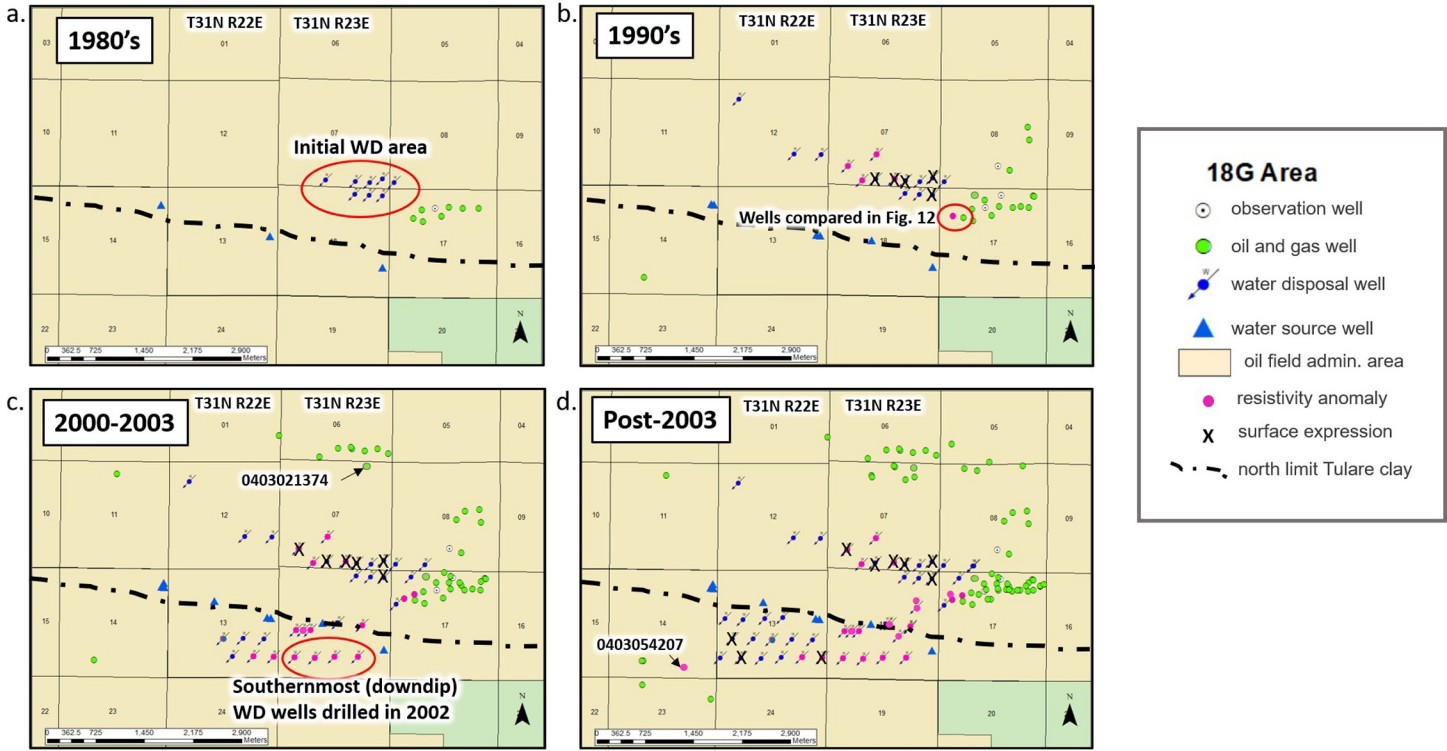

**Fig 13. Post-1980 wells drilled in the 18G WD area on the south flank of Elk Hills Oil Field.** The Tulare Formation dips to the south. Wells drilled during this time period but not logged through the Tulare Formation are not shown. Northern limit of Tulare clay (outcrop) shown as black dot-dash line (from California Resources Corporation [42]). a) wells drilled in the 1980's with initial 18G WD area in red circle b) additional wells drilled in the 1990's. Wells highlighted in pink show resistivity anomalies indicating the presence of saline produced water in the Tulare Formation. X's mark the locations where injected water has broken through to the surface (surface expressions). Water supply wells (blue triangles) were sampled multiple times during the 1990's [41] but showed no effects of disposal water migration. No samples are available for the post-1990's period. c) wells drilled from 2000 to 2003. Well 0403021374 contains perched brackish water in a sand directly above the Amnicola clay and lies updip from the 18G disposal wells. However, this brackish perched water is also present in wells dating back to the 1940's and does not appear to be related to water disposal in the 18G area. d) wells drilled post-2003. Well 0403054207 was drilled in 2014 and resistivity anomalies indicate that disposal water had migrated beyond the boundary of the Elk Hills Oil Field into the Buena Vista Oil Field at that time.

In the early 2000s, disposal operations shifted farther south (downdip) to the south boundary of the Elk Hills Oil Field where injection could be confined below the Tulare clay (Fig 13C) to mitigate the development of surface expressions that occurred in the 1990s in areas where the Tulare clay was not present although surface expressions continued to occur north of the Tulare clay outcrop during this time. Four of the wells drilled along the southern oil field boundary in April and September 2002 showed saline resistivity anomalies (southernmost WD wells noted in Fig 13C). Resistivity anomalies in one of these wells (0403020256) are highlighted in pink in the cross section in Fig 11. By this time, the plume had moved south (downdip) beyond the water source wells. It seems likely that the resistivity anomalies in the four wells along the southern boundary of the oil field are due to southward migration of the plume from the original WD area in the 1980's—a distance of approximately 1,200 m (4,000 ft) because only three WD wells were injecting produced water between the southernmost wells and the original WD area prior to 2002. These wells began injecting in mid-June and late November 2001, while the southern boundary wells were drilled in April and September 2002 making it unlikely that the 2001 WD disposal wells were the sole source of the resistivity anomalies in the 2002 wells. In addition, the logs from two of these wells—0403019381 (logged November 30, 2001) and 0403041542 (logged March 2002)—already displayed resistivity anomalies.

To the north (updip) a single sand within the vadose zone at 120 m (400 ft) deep in a well logged in November 2002 (API 0403021374) contains brackish water (about 3.2 ohm-m resistivity) perched upon the Amnicola clay layer about 1,430 m (4,700 ft) from the disposal field (Fig 13C). However, this perched brackish water is present in logs in adjacent wells drilled as early as the late 1940's (pre-dating disposal by injection in the 18G area) and appears to be a natural feature of the aquifer.

From 2003 to 2005, injection wells were drilled farther west along the southern border of the Elk Hills Oil Field, and a series of surface expressions occurred in Section 13 from 2006 to 2015 even though injection in the southwest part of the disposal area occurred below the Tulare clay (Fig 13D). A well (0403054207) drilled in 2014, 550 m (1,800 ft) west of the 18G area in the adjacent Buena Vista Oil Field, has resistivity anomalies that appear to be related to the disposal activities in the 18G area in Elk Hills Oil Field (Fig 13D).

In the 23/25Z area in western Elk Hills and eastern Asphalto Oil Fields (Figs 4A and S7), disposal originally occurred in 1981–82 within the upper Tulare Formation in an area where the Tulare clay was not present (S8 Fig). Additional WD injectors in this area were drilled between 1983 and 1986. After two surface expressions in 1989 and 1992 [51] (S8 Fig), further disposal shifted westward from 2001 to 2014 and occurred within the upper Tulare Formation below the Tulare clay and in the lower Tulare Formation below the Amnicola clay (S8 Fig). In 2014, disposal shifted to the southeast where, as of this writing, saline water was being injected into the vadose zone below the Amnicola clay on the western end of the anticline crest at Elk Hills (S8 Fig).

Upper Tulare Formation disposal is being phased out at Elk Hills Oil Field, and most of the produced water injection since 2004 occurs within the vadose zone below the Amnicola clay along the crest of the Elk Hills anticline in the 27R disposal area (Figs 4A and S9) providing an artificial source of aquifer recharge to the lower Tulare Formation. No surface expressions have been observed associated with injection into the vadose zone to date. However, some migration of saline injected fluids through previously unsaturated sands in the vadose zone within the lower Tulare Formation has been observed in geophysical logs from newer wells drilled in the northwestern part of the 27R area from 2007 to 2014 [52] (S10 Fig). These perched saline sands along the crest of the Elk Hills anticline in the 27R WD area are shown highlighted in pink in the cross section in Fig 11 (wells 0403045666 and 0403035654).

## Discussion and conclusions

As water shortages become more frequent—especially in the arid southwestern US—desalination of brackish aquifers (TDS less than 10,000 mg/L TDS) for beneficial use is becoming more attractive. This is especially true in California's San Joaquin Valley which contains the three largest agricultural-producing counties in the US (Fresno, Kern, and Tulare Counties respectively - https://thebusinessjournal.com/central-valley-tops-list-of-u-s-ag-counties/; accessed 11/3/2021).

Kern County, the location of our study area, was the largest crude oil producing county in the US as of 2011 (https://www.ers.usda.gov/data-products/county-level-oil-and-gas-production-in-the-us/; accessed 11/3/2021), but many of the oil fields in Kern County are 100 years old and in decline. As oil fields age, the quantity of water co-produced with the oil and gas tends to increase and this water is often saline. Increased oil production due to hydraulic fracturing often increases the volume of produced water as well—creating greater disposal issues. Today much of the produced water is injected into brackish aquifers via disposal wells.

The results of our study show that geophysical log analysis can be a viable substitute for water sampling in order to determine the distribution of brackish water aquifers. This finding is important because most water wells are not completed in the poor-quality brackish water aquifers and sampling opportunities are rare. Use of these methods can provide regulatory agencies with information needed to determine which aquifers require protection under federal and state laws. As additional wells are drilled within oil fields to replace wells with mechanical issues or to expand production and disposal projects, the comparison between the geophysical logs in the new and old wells show how operations in the oil fields have impacted the adjacent aquifers and which aquifers are likely to be impacted in the future.

Geophysical log calculations of salinity and laboratory analysis of water samples from oil and water-source wells completed in the Tulare aquifer in the study area show that the upper part of the Tulare aquifer (above the Amnicola clay) contains brackish formation water with salinity less than 10,000 mg/L TDS at the margins of the Elk Hills anticline. Salinity increases with depth and the transition to water with TDS greater than 10,000 mg/L (base 10K) typically occurs within the lower part of the Tulare aquifer between the Amnicola clay and the base of the Tulare Formation (Fig 11). As a result, the depth to base 10K is shallower along the crests of the anticlines and deeper within the adjacent synclines such as the Buena Vista Valley (Fig 9). Because the Tulare Formation is thinner above the anticlinal crests (Fig 5), the thickness of the aquifer containing water with less than 10,000 mg/L TDS is also less in these areas and, consequently, thicker in the adjacent synclines (Figs 10 and 11). In many areas along the crest of the Elk Hills anticline, nearly the entire Tulare Formation lies above the water table, resulting in a thick vadose zone (Fig 11). The higher elevation of the base of the Tulare Formation along the anticline crest results in high water-table elevations at Elk Hills relative to the surrounding area (Fig 7) indicating potential lateral groundwater head gradients towards adjacent groundwater basins where groundwater is used for supply.

The structural relationship between the depth to base 10K and the elevation of the base of the Tulare Formation breaks down in the easternmost part of the study area in the Coles Levee Oil Fields. Here the depth to Base 10K becomes progressively deeper to the east even as the base of the Tulare Formation begins to climb eastward onto the Bakersfield Arch (Figs 3 and 9). This area receives substantial freshwater recharge from the Kern River and contains numerous irrigation and domestic water supply wells as well as the Kern Water Bank.

Groundwater salinity in the Tulare aquifer in the western part of the study area is affected by injection of produced water that has a much higher salinity than the brackish Tulare Formation water. The effect of the produced water injection is interpreted in geophysical logs as

resistivity anomalies such that the affected aquifer sands have much lower resistivity (higher salinity) than underlying sands, causing a local reversal in the typical vertical salinity gradient of increasing salinity with depth (Figs 8 and 12A). In some cases, only the lower parts of the aquifer sand layers are affected causing these sands to exhibit high resistivity near the top and low resistivity near the bottom, as denser saline water tends to flow along the base of the sand layers (Fig 12B). Density neutron logs are often affected by injected produced water as well and sands containing produced water exhibit numerous intervals approximately one meter (three feet) thick in which the neutron curve reads a lower porosity than the density curve—a behavior termed cross-over (S6 Fig). Cross-over typically indicates the presence of gas in the sand and may represent methane created by bacterial metabolism of organic carbon in the injected water or thermogenic gases present in the injected water.

An analysis of resistivity anomalies in wells drilled and logged at various time intervals post-injection in the 18G water disposal area on the south flank of the Elk Hills Oil Field indicate that the injected water migrated downdip (south) from the disposal area toward the southern oil field boundary for a distance of approximately 1,200 m (4,000 ft) from 1981 to 2002 (Fig 13). The downdip migration from 1981 to 2002 was likely aided by drawdown from oil field water-source wells located south of the initial disposal wells. However, by 2002, wells drilled and logged south of the water-source wells indicated that the saline water had spread beyond the water-source wells downdip to the edge of the oil field (Fig 13C). Shincariol and Schwartz [64] found that, when injecting saline water into dipping layers containing fresh water in experimental tanks, the density forces provide a component of flow down the topographic gradient of the interface (i.e., down the dip of the layers) similar to our observations of the southward movement of saline injected fluids on the south flank of Elk Hills anticline.

Unfortunately, no post-2002 wells are located south of the oil field boundary to determine the extent of the saline injected water in this direction.

Current produced water disposal practices at Elk Hills Oil Field have shifted northward toward the crest of the Elk Hills anticline into the unsaturated sands of the lower Tulare Formation in the 27R disposal area and in parts of the 23/25Z area in the western Elk Hills and Asphalto Oil Fields providing a source of artificial recharge to the lower Tulare aquifer along the crest of Elk Hills. The injected water appears to be migrating to the north in the 27R area as evidenced by geophysical logs in newer wells in that area that show saline water filling previously unsaturated sands in the lower Tulare Formation [52].

## Supporting information

**S1 File. Method for generating the Water Surface Elevation (WSE) maps [66, 67].** (DOCX)

**S1 Table. Wells with surface expressions shown in Fig 4B.** Date 1 is the date of the first surface expression in the well and Date2 is the date of the second surface expression (if any). Data from well history files available from CalGEM's online Well Files [51]. (TIF)

**S2 Table. Fitted parameters for the Gaussian process used to interpolate Water Surface Elevation (WSE).** (TIF)

**S1 Fig. Geophysical log showing example of Tulare Formation correlation picks used in map generation.** Left track contains GR = Gamma Ray log in red, SP = spontaneous potential log in black and right track contains shallow resistivity log in black. The yellow interval is the Tulare clay and the blue interval is the Amnicola clay. The Amnicola clay divides the upper

and lower Tulare Formation for this study. Well location shown in Fig 3.
(TIF)

**S2 Fig. Two examples showing contrasts in geophysical log response at the boundary between the vadose zone and the water table.** Negative SP is shaded yellow and resistivity curve is shaded yellow at values greater than three ohms. Resistivity readings are typically higher in unsaturated sands above the water table—particularly if the groundwater is brackish. Density-neutron log is shaded yellow in areas where the neutron porosity is lower than the density—a phenomenon known as cross-over that indicates the presence of air or natural gas. The top of the water table is picked at the base of cross-over in the density-neutron log. The density-neutron log is shaded brown in intervals where the neutron curve reads higher porosity than the density curve—the larger the difference, the greater the volume of clay within the formation. Clean sand intervals typically have little to no separation between the curves.
(TIF)

**S3 Fig. Flow charts depicting the process used to determine Water Surface Elevation (WSE) maps.** a) A probabilistic generative model of WSE data, as described in S1 File and b) an equivalent model, used for computing model likelihood. c) Flow chart showing steps used to calculate salinity from geophysical logs in Fig 8 and construction of the Base 10K map in Fig 9.
(TIF)

**S4 Fig. Cross section through the Buena Vista lakebed depicting the relationship between the Tulare clay (yellow) and Corcoran Clay (green).** Corcoran Clay picks based on Croft's [45] E-clay picks in wells 31S25E27F01 and 0402938056. Tulare clay picks from this study (note that SP curve is reversed in 31S25E27F01). Air photo base on inset map from https://basemap.nationalmap.gov/arcgis/rest/services/USGSImageryOnly/MapServer/.
(TIF)

**S5 Fig. The Gaussian process model used to create the Water Surface Elevation (WSE) maps.** These maps provide a probability distribution for WSE at each location; therefore, we can quantify the uncertainty associated with each WSE prediction. At each location, the model provides a mean and variance, taking the square root of the variance gives the standard deviation ($\sigma$). The $\sigma$ maps should be considered with the WSE maps (Fig 7) to understand which zones have higher or lower uncertainty. The uncertainty estimates are a function of distance from the input data. Uncertainty is lower when a WSE prediction is near a water table measurement (in space or time), conversely, predictions farther from input data have higher associated uncertainty. For example, on S5b Fig (1990) high uncertainties indicate input data are rather limited in the Elk Hills Oil Field which creates a data gap with the adjacent groundwater basin to the northeast. Therefore, caution should be used when interpreting the WSE predictions in the corresponding location on the WSE map (Fig 7B). By 2000 and 2010 more data are available in eastern Elk Hills and the uncertainty lowers. Basemap from Esri. "World Hillshade" [basemap]. Scale Not Given. "World Hillshade". July 9, 2015. https://www.arcgis.com/home/item.html?id=1b243539f4514b6ba35e7d995890db1d. (September 30, 2021).
(TIF)

**S6 Fig. A comparison of two nearby wells–one drilled after two years of produced water disposal by injection in the 18G area of Elk Hills (1982 well) and one after 15 years of water disposal (1995 well) (locations shown on Fig 13B and resistivity logs shown in Fig 12B).** Abnormally low resistivity in the sands are highlighted in pink in the 1995 well. These intervals indicate the presence of saline produced water in the sands of the Tulare aquifer. The 1995 well also exhibits small intervals (about one meter (3 ft)) of density-neutron cross-over

(shaded in yellow and highlighted by red circles) in and near the affected sands. This density-neutron behavior is common in sands affected by invasion of oilfield produced water and may indicate bacterial metabolism of traces of organic matter in the injected water and the consequent presence of methane in the sands.
(TIF)

**S7 Fig.** a) Type log from the 23Z/25Z water disposal (WD) project showing depth to water table from density-neutron cross-over, major clay units and completion interval. b) map showing location of the 23Z/25Z WD area in western Elk Hills (yellow box). Contours show elevation on top of the base of the Tulare Formation measured in meters with respect to mean sea level. Photographic base from https://basemap.nationalmap.gov/arcgis/rest/services/USGSImageryOnly/MapServer.
(TIF)

**S8 Fig. Effects of produced water disposal in the 23Z/25Z area.** a) Example of log-calculated TDS vs depth showing effects of water disposal (WD) on salinity profile. b) Map of 23Z/25Z WD area showing the locus of produced water disposal in various years. The location and year of resistivity anomalies (red stars) and surface expressions (purple squares) caused by disposal of saline produced water into the Tulare Formation are also shown.
(TIF)

**S9 Fig.** a) Type log of the Tulare Formation in the 27R WD area showing vadose zone from density-neutron logs, major clay layers and completion interval. b) Location of the 27R water disposal project area in western Elk Hills (yellow box). Contours show elevation on top of the base of the Tulare Formation measured in meters with respect to mean sea level. Photographic base from https://basemap.nationalmap.gov/arcgis/rest/services/USGSImageryOnly/MapServer.
(TIF)

**S10 Fig. Map of 27R WD area with red circles showing the locus of produced water disposal in various years.** Water disposal wells are shown in blue, Enhanced oil recovery injection wells in black, gas producing wells in red and oil producing wells in green. The location and year of resistivity anomalies (red stars) caused by disposal of saline produced water into the Tulare Formation are also shown. In this case, the resistivity anomalies are noted by the fill-up of previously unsaturated sands with produced water creating saline perched aquifers above the regional water table.
(TIF)

## Acknowledgments

This work is a product of the California State Water Resources Control Boards Oil and Gas Regional Monitoring Program. Raster logs were digitized by Dr. David Shimabukuro and Theron Sower's team of geology students from California State University Sacramento. The appearance of the maps in the figures were greatly improved by the U.S. Geological Survey Sacramento Publishing Service Center. Any use of trade, firm, or product names is for descriptive purposes only and does not imply endorsement by the U.S. Government.

## Author Contributions

**Conceptualization:** Janice M. Gillespie.

**Data curation:** Janice M. Gillespie, Michael J. Stephens, John G. Warden.

**Formal analysis:** Janice M. Gillespie, Michael J. Stephens, John G. Warden.

**Investigation:** Janice M. Gillespie, Michael J. Stephens, Will Chang, John G. Warden.

**Methodology:** Janice M. Gillespie, Michael J. Stephens, Will Chang, John G. Warden.

**Software:** Michael J. Stephens, Will Chang.

**Validation:** Janice M. Gillespie.

**Visualization:** Janice M. Gillespie.

**Writing – original draft:** Janice M. Gillespie.

**Writing – review & editing:** Michael J. Stephens, John G. Warden.

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
