## [Decision Letter · Decision Letter 0]

4 Oct 2021

PONE-D-21-28543Mapping aquifer salinity gradients and effects of oil field produced water disposal using geophysical logs:  Elk Hills, Buena Vista and Coles Levee Oil Fields, San Joaquin Valley, CaliforniaPLOS ONE

Dear Dr. GILLESPIE,

Thank you for submitting your manuscript to PLOS ONE. After careful consideration, we feel that it has merit but does not fully meet PLOS ONE’s publication criteria as it currently stands. Therefore, we invite you to submit a revised version of the manuscript that addresses the points raised during the review process.

We look forward to receiving your revised manuscript.

Kind regards,

Zaher Mundher Yaseen

Academic Editor

PLOS ONE

“This work was primarily funded by the California State Water Resources Control Board Oil and Gas Regional Monitoring Program and supplemental US Geological Survey Cooperative Matching Funds.  Advice and editing from Matt Landon, Lyndsay Ball and Pete McMahon of the US Geological Survey are greatly appreciated.  Raster logs were digitized by David Shimabukuro and Theron Sower’s team of geology students from California State University Sacramento.  The appearance of the maps in the figures were greatly improved by the US Geological Survey Sacramento Publishing Service Center.”

“This work was primarily funded by the California State Water Resources Control Board Oil and Gas Regional Monitoring Program and supplemental US Geological Survey Cooperative Matching Funds.  The funders had no role in study design, data collection and analysis, decision to publish, or preparation of the manuscript.”

“NO”

5. We note that Figures 1, 3, 4, 5, 6, 7, 9, 10, 11, 13, S3,S4, S6-S9 in your submission contain [map/satellite] images which may be copyrighted. All PLOS content is published under the Creative Commons Attribution License (CC BY 4.0), which means that the manuscript, images, and Supporting Information files will be freely available online, and any third party is permitted to access, download, copy, distribute, and use these materials in any way, even commercially, with proper attribution. For these reasons, we cannot publish previously copyrighted maps or satellite images created using proprietary data, such as Google software (Google Maps, Street View, and Earth). For more information, see our copyright guidelines: http://journals.plos.org/plosone/s/licenses-and-copyright.

 a. You may seek permission from the original copyright holder of Figure 1, 3, 4, 5, 6, 7, 9, 10, 11, 13, S3,S4, S6-S9 to publish the content specifically under the CC BY 4.0 license. 

Additional Editor Comments (if provided):

Reviewers' comments:

Reviewer's Responses to Questions

**Comments to the Author**

1. Is the manuscript technically sound, and do the data support the conclusions?

Reviewer #1: Yes

Reviewer #2: Yes

2. Has the statistical analysis been performed appropriately and rigorously? 

Reviewer #1: Yes

Reviewer #2: I Don't Know

3. Have the authors made all data underlying the findings in their manuscript fully available?

Reviewer #1: Yes

Reviewer #2: Yes

4. Is the manuscript presented in an intelligible fashion and written in standard English?

Reviewer #1: Yes

Reviewer #2: Yes

5. Review Comments to the Author

Reviewer #1: The authors reported an excellent research on the investigation of Mapping aquifer salinity gradients and effects of oil field produced water disposal using geophysical logs. The manuscript is within the journal scope and has the potential to be accepted. However, there are several comments to be resolved before the manuscript publication.

1- The abstract is too lengthy, it shall be revised to present only the essential information and in one single paragraph.

2- All abbreviations shall be defined once you have mentioned at first. Example TDS, etc.

3- Modeling flowchart should be included in the method section for better understanding by the readers.

4- As per the overlapping of the manuscript: 8% was observed related to the following document

https://archives.datapages.com/data/deg/2019/EG032019/eg18009/eg18009.html

Authors are responsible to reduce the plagiarism percentage for their ethical right.

5- A new section presenting the practical aspect of the attained results shall be reported in the manuscript.

6- English editing is recommended for the article.

Reviewer #2: The manuscript “Mapping aquifer salinity gradients and effects of oil field produced water disposal using geophysical logs: Elk Hills, Buena Vista and Coles Levee Oil Fields, San Joaquin Valley, California”; is technically sound, and data support the conclusions. The conclusions consistent with the evidence and arguments presented. Although it is relevant, interesting and well written, but it is not easy to read. Tables and figures aid understanding, but I think, author used huge number of figures, and some of them are superfluous. However, However, many notes listed below have emerged that need to be addressed, and without they would not be accepted for publication. Consequently, I recommend a major revision.

1. In order to complete the elements of the introduction, the previous studies must be mentioned, so I find that some sentences in the introduction need to be reformulated to show the problem of the research and its importance clearly, and what is the new addition (novelty) to the field of knowledge after listing the relevant conclusions of previous researches. At the end of this report, I have put a number of useful links that can be quoted to support some ideas and scientific facts.

2. Line 57: Please delete “milligrams per liter”, It is enough to be mentioned by “mg/l”.

3. Line 77: What do you mean by “native waters”, do it mean fresh water, if yes, please use the last term.

4. Line 83-84: The Elk Hills Oil Field (formerly Naval Petroleum Reserve No.1) has produced over one billion barrels (160,000,000 m3) of oil; Please fix the period of time (year) during which that huge amount of oil was produced.

5. Fig. 1: The red line represented the San Andreas fault zone appears to be extended somewhat outside the map boundary. Please correct.

6. Fig. 2: There are four rectangles that appear as blanks of white color! What do they represent? Could you add the thickness of the Tulare aquifer, please? There are three beds of clay. Is aquifer a multi-confined aquifer. What is a water-bearing lithology?.

7. Line 108: The abbreviation “SJV forms” comes suddenly, please define it at the first appearance.

8. Fig.3: This fig is of low resolution.

9. Line 180: Please, it is enough to write 23 cm) (5 to 9 in, and not necessary to write in detail as 23 centimeters (cm) (5 to 9 inches (in)).

10. Line 182: The same comment above for “centimeters per year (cm/yr) (6 inches per year (in/yr)).

11. Lines 212-213: “The contact between the Tulare and the overlying alluvium was placed at a depth of 144 m (473 ft) in well 02938955 in the Buena Vista Valley”; where is the location on map of this well?.

12. For the better presentation of the hydrological system, you need a 3D model for the aquifer.

13. Caption of Table 1 should be placed above (not bellow) table.

14. Method section: It has been written very briefly; I would prefer to rewrite it in details.

15. The date of water sampling is very important to be mentioned, particularly for the water quality.

16. Line 347-348; the mention to the figures should be sequentially, here in the statement “Geophysical well log correlations were used to map the elevation of the base of the Tulare Formation (Fig 3) and its thickness (Fig 5)”, author has mentioned FIG 5 Before mentioning to Fig 4.

17. Line 364: the site cannot be reached.

18. One of the weaknesses of the methods is the lack of regularly distributed wells. The other thing is determining the net clay thickness without using core data.

19. Why was the figure numbering system changed to S1-S9 after Fig 13?

20. Line 702: Caption of Table 2 must be above the table not bellow.

21. Line 754-755: “This suggests that the saline injected water and brackish formation waters are segregated by density and mixing has not had time to occur in some sands”. It is very difficult to believe this suggestion for the simple reason that salinity is represented by positive and negative ions and these ions can move quickly in the solution, which quickly becomes homogeneous with time and after a short period

22. Many useful papers can be cited as suitable, Introductas:

- https://link.springer.com/article/10.1007/s11053-021-09923-4

- https://www.sciencedirect.com/science/article/abs/pii/S0264817220304207

- https://link.springer.com/article/10.1007/s13201-019-0944-6

- https://link.springer.com/chapter/10.1007/978-3-030-01572-5_12

- https://jprs.gov.iq/index.php/jprs/article/view/273

- https://link.springer.com/article/10.1007/s12517-018-3908-5

- https://link.springer.com/article/10.1007/s40808-018-0510-5

-

6. PLOS authors have the option to publish the peer review history of their article (what does this mean?). If published, this will include your full peer review and any attached files.

Reviewer #1: No

Reviewer #2: **Yes: **Salih Muhammad Awadh

---

## [Author Response · Author response to Decision Letter 0]

6 Jan 2022

This information is supplied in the response to reviewers file which has been uploaded to the editorial manager in a previous section. Yellow highlights in MS with tracked changes are flagged areas of overlap with Gillespie et al. 2019

---

## [Decision Letter · Decision Letter 1]

20 Jan 2022

Mapping aquifer salinity gradients and effects of oil field produced water disposal using geophysical logs:  Elk Hills, Buena Vista and Coles Levee Oil Fields, San Joaquin Valley, California

PONE-D-21-28543R1

Dear Dr. GILLESPIE,

We’re pleased to inform you that your manuscript has been judged scientifically suitable for publication and will be formally accepted for publication once it meets all outstanding technical requirements.

Kind regards,

Zaher Mundher Yaseen

Academic Editor

PLOS ONE

Additional Editor Comments (optional):

Reviewers' comments:

Reviewer's Responses to Questions

**Comments to the Author**

1. If the authors have adequately addressed your comments raised in a previous round of review and you feel that this manuscript is now acceptable for publication, you may indicate that here to bypass the “Comments to the Author” section, enter your conflict of interest statement in the “Confidential to Editor” section, and submit your "Accept" recommendation.

Reviewer #2: All comments have been addressed

2. Is the manuscript technically sound, and do the data support the conclusions?

Reviewer #2: Yes

3. Has the statistical analysis been performed appropriately and rigorously? 

Reviewer #2: N/A

4. Have the authors made all data underlying the findings in their manuscript fully available?

Reviewer #2: Yes

5. Is the manuscript presented in an intelligible fashion and written in standard English?

Reviewer #2: Yes

6. Review Comments to the Author

Reviewer #2: The author has responded to most of the reviewer's comments, so the manuscript looks better than it was in the initial version. In my opinion, the manuscript has become acceptable for publication in your journal.

7. PLOS authors have the option to publish the peer review history of their article (what does this mean?). If published, this will include your full peer review and any attached files.

Reviewer #2: **Yes: **Salih Muhammad Awadh

---

## [Editor Report · Acceptance letter]

10 Mar 2022

PONE-D-21-28543R1 

Mapping aquifer salinity gradients and effects of oil field produced water disposal using geophysical logs:  Elk Hills, Buena Vista and Coles Levee Oil Fields, San Joaquin Valley, California 

Dear Dr. Gillespie:

I'm pleased to inform you that your manuscript has been deemed suitable for publication in PLOS ONE. Congratulations! Your manuscript is now with our production department. 

Kind regards, 

on behalf of

Dr. Zaher Mundher Yaseen 

Academic Editor

PLOS ONE